# Resistance to targeted therapies as a multifactorial, gradual adaptation to inhibitor specific selective pressures

Robert Vander Velde [1,2], Nara Yoon[3], Viktoriya Marusyk[1], Arda Durmaz[3,4], Andrew Dhawan [3], Daria Miroshnychenko [1], Diego Lozano-Peral [1,5], Bina Desai [1,6], Olena Balynska[1], Jan Poleszhuk [7], Liu Kenian[8], Mingxiang Teng [9], Mohamed Abazeed[3], Omar Mian[3], Aik Choon Tan[9], Eric Haura[10], Jacob Scott [3,4✉] & Andriy Marusyk [1,2✉]

Despite high initial efficacy, targeted therapies eventually fail in advanced cancers, as tumors develop resistance and relapse. In contrast to the substantial body of research on the molecular mechanisms of resistance, understanding of how resistance evolves remains limited. Using an experimental model of ALK positive NSCLC, we explored the evolution of resistance to different clinical ALK inhibitors. We found that resistance can originate from heterogeneous, weakly resistant subpopulations with variable sensitivity to different ALK inhibitors. Instead of the commonly assumed stochastic single hit (epi) mutational transition, or drug-induced reprogramming, we found evidence for a hybrid scenario involving the gradual, multifactorial adaptation to the inhibitors through acquisition of multiple cooperating genetic and epigenetic adaptive changes. Additionally, we found that during this adaptation tumor cells might present unique, temporally restricted collateral sensitivities, absent in therapy naïve or fully resistant cells, suggesting the potential for new therapeutic interventions, directed against evolving resistance.

[1] Department of Cancer Physiology, H Lee Moffitt Cancer Centre and Research Institute, Tampa, FL, USA. [2] Department of Molecular Medicine, University of South Florida, Tampa, FL, USA. [3] Department of Translational Hematology and Oncology Research, Cleveland Clinic, Cleveland, OH, USA. [4] Systems Biology and Bioinformatics, Case Western Reserve University School of Medicine, Cleveland, OH, USA. [5] Supercomputer and Bioinnovation Center, University of Málaga, Málaga, Spain. [6] University of South Florida Cancer Biology PhD Program, Tampa, FL, USA. [7] Nalecz Institute of Biocybernetics and Biomedical Engineering, Polish Academy of Sciences, Warsaw, Poland. [8] Department of Pathology, H Lee Moffitt Cancer Centre and Research Institute, Tampa, FL, USA. [9] Department of Biostatistic and Bioinformatics, H Lee Moffitt Cancer Centre and Research Institute, Tampa, FL, USA. [10] Department of Thoracic Oncology, H Lee Moffitt Cancer Centre and Research Institute, Tampa, FL, USA. ✉email: scottj10@ccf.org; Andriy.Marusyk@moffitt.org

Despite inducing strong clinical responses, inhibitors that target abnormal activities of oncogenic tyrosine kinases (TKIs), including those directed against oncogenic ALK signaling in non-small cell lung cancers (NSCLCs), are rarely curative in advanced disease[1]. As ALK-TKIs typically fail to eradicate all of the tumor cells, residual tumors eventually acquire resistance and relapse.

In contrast to the large body of work on deciphering the molecular mechanisms of resistance, understanding of its evolutionary mechanisms is more limited. Based on quantitative analyses and the detection of resistance-associated mutations, resistance is often assumed to arise due to selective expansion of pre-existent rare, fully resistant subpopulations[2–5]. On the other hand, a growing body of experimental studies suggest that resistance can emerge de novo from drug-tolerant persister (DTP) cells, which can maintain residual disease and serve as a substrate for mutational or epigenetic conversions to fully resistant phenotypes[6]. Finally, as therapies can induce adaptive phenotypic changes on shorter time scales, acquired resistance has also been viewed from a differentiation/reprograming paradigm[7–9].

Motivated by interest in evolutionary-informed therapy scheduling, we investigated the origin and dynamics that underlie the development of therapy resistance of an in vitro model of ALK + NSCLC, the patient-derived NCI-H3122 cell line. Upon exposure to different ALK-TKIs, these cells rapidly and predictably develop strong drug resistance. Resistance originates de novo, from weakly resistant heterogeneous subpopulations, which differ in fitness when exposed to different ALK-TKIs. Levels of resistance gradually increase under therapy, through acquisition of multiple cooperating genetic and epigenetic mechanisms, through TKI-specific phenotypic evolutionary trajectories. In contrast to therapy-naive or fully resistant cells, these evolving populations show strong collateral sensitivity to the dual epidermal growth factor receptor EGFR/HER2 inhibitor lapatinib, suggesting a temporally restricted opportunity to interfere with the development of resistance.

## Results

**Different ALK-TKIs select for distinct resistant phenotypes.** To understand the evolution of resistance to different TKIs, we focused on the ALK + NSCLC cell line, NCI-H3122, which has been well-characterized in multiple mechanistic studies. Starting from resistant cell lines derived through a dose-escalation protocol in our previous study[10], we continued dose escalation, eventually selecting for cells capable of growing in high, clinically relevant concentrations of the drugs (up to 1 µM crizotinib, 4 µM lorlatinib, 2 µM alectinib, and 200 nM ceritinib). Cells with evolved resistance to ALK-TKI (erALK-TKI) displayed strong collateral resistance toward other ALK-TKIs, with IC50's 5–100× higher than therapy naïve controls (Fig. 1a and Supplementary Table 1). Despite the similar shift in IC50, erALK-TKI cells selected by different inhibitors displayed divergent responses to higher drug concentrations. Consistent with the clinical efficacy of alectinib and lorlatinib as a second-line therapy after failure of crizotinib[11,12], high concentrations of alectinib and lorlatinib strongly inhibited cells with evolved resistance to crizotinib (erCriz) and ceritinib (erCer). The resistance was partially or completely maintained after drug holiday both in vitro (Supplementary Fig. 1a) and in vivo (Fig. 1b, c). Resistance was not an artifact of the dose-escalation protocols, as H3122 cells developed compatible resistance levels after 2–4 months of acute exposure to clinically relevant concentrations of the ALK-TKIs (Fig. 1d). Similar to the resistant cell lines derived by gradual exposure, resistant phenotypes in cell lines derived by acute exposure to ALK-TKIs were largely heritable (Supplementary Fig. 1b). Next,

we asked whether the differences in cross-sensitivities of erALK-TKI cells towards different ALK inhibitors are attributable to the choice of specific ALK inhibitor. To this end, we examined the sensitivities of three independently derived (acute exposure protocol) cell lines for each of the ALK-TKIs used. Consistent with previous findings (Fig. 1a), erLor and erAlec cells demonstrated stronger resistance to high concentrations of different ALK-TKIs, compared with erCriz and erCer cell lines (Fig. 1e).

Next, we asked whether the differences in ALK-TKI cross-sensitivity of cells selected with different ALK-TKIs are associated with specific changes in EML4-ALK-dependent signaling pathways. We found that, in the absence of ALK-TKI, all of the erCriz cell lines displayed increased phosphorylation of EML4-ALK, which was in most cases associated with elevated total EML4-ALK protein levels, as well as increased STAT3 phosphorylation (Fig. 1f). In contrast, all of the erLor cell lines displayed reduced EML4-ALK and STAT3 phosphorylation. Instead, they expressed higher levels of EGFR, HER2, or both. To assess the phenotypes more globally, we examined mRNA expression levels of 230 cancer-related genes using the Nanostring nCounter GX human cancer reference panel. Principal component analyses (PCAs) and hierarchical clustering for mRNA expression were largely consistent with the immunoblot evaluation of phosphorylation (Supplementary Fig. 2a, b). erCer, evolved through acute selection, as well as acutely and gradually evolved erCriz and erLor cells formed distinct clusters; in contrast, phenotypes of erAlec were more diverse.

To test whether the observed differences reflected different proportions of shared phenotypic subpopulations or population-wide phenotypic changes, we performed single-cell RNA sequencing of erALK-TKI cell lines, focusing on two cell lines per specific ALK-TKI, with the highest divergence in PCA analysis of NanoString data (Supplementary Fig. 2a). Uniform manifold approximation and projection (UMAP)[13] dimension reduction of single-cell expression data revealed relative phenotypic homogeneity within individual erALK-TKI lines, with a high degree of similarity among cell lines derived with the same inhibitor (Fig. 1g). Thus, despite a degree of stochasticity, reflective of evolutionary contingencies[14], acquired resistance to specific ALK-TKIs is associated with phenotypes that are convergent within the same inhibitor, but divergent between different inhibitors.

**Resistance originates from diverse tolerant subpopulations.** Acquired resistance is often assumed to reflect a simple expansion of therapy-resistant subpopulations[15,16]. Whereas the observed predictable distinctions between resistant phenotypes, selected in different ALK-TKIs contradicts this notion, we decided to interrogate the pre-existence of fully resistant phenotypes more directly. To this end, we seeded treatment-naive NCI-H3122 cells at clonogenic densities in the presence of ALK-TKIs or vehicle control (dimethylsulfoxide (DMSO)). At 10 days post seeding, the majority of treatment-naive cells plated in DMSO and erALK-TKI cells plated in DMSO or ALK-TKIs formed macroscopic colonies. In contrast, in the presence of ALK-TKIs, treatment-naive cells formed only microscopic colonies, consistent with tolerance, even upon plating as many as 10,000 cells (Fig. 2a). At the same time, limiting dilution experiments revealed that 1:338–1:660 of treatment-naive NCI-H3122 cells can give rise to robustly growing drug-resistant colonies within 10 weeks of drug exposure (Fig. 2b–d). Whereas these observations cannot exclude pre-existence of rare (<0.01%) fully resistant subpopulations, together with the observed inhibitor-specific divergence of resistance phenotypes, they indicate that acquisition of resistance de novo via tolerant intermediates must be more common in our

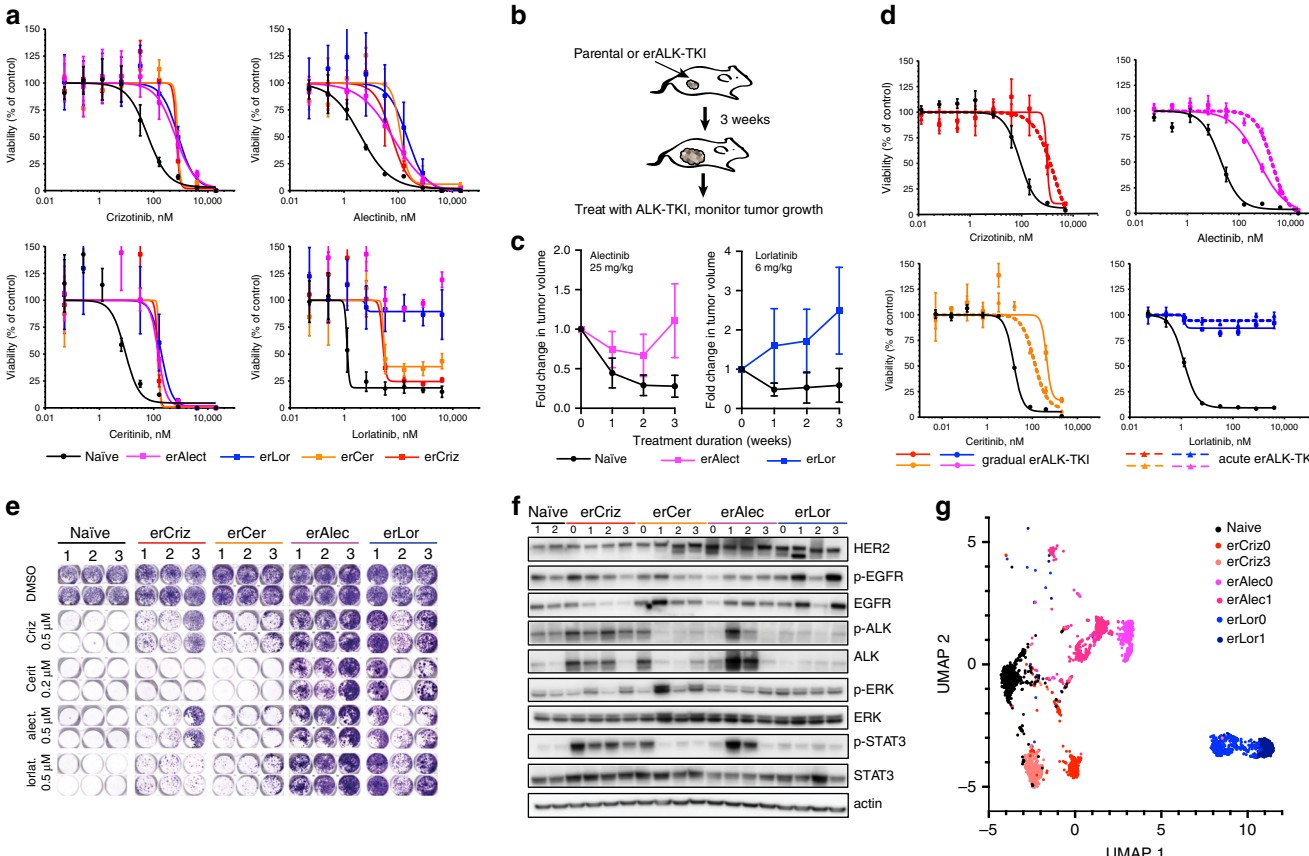

**Fig. 1 Characterization of evolved resistance to ALK-TKIs. a** Sensitivities of the treatment-naive H3122 cell line and its erALK-TKI derivative cell lines to the indicated ALK inhibitors, measured by Cell Titer Glo assay. Y-axis represents luminescent signal normalized to DMSO control. Mean ± SD of experimental triplicates are shown. **b** Experimental schemata of the xenograft experiment. Treatment is initiated after macroscopic tumors are formed within 3 weeks in the absence of therapies. **c** Growth dynamics of xenograft tumors initiated by inoculation of erAlect or erLor cells under treatment with indicated doses of ALK-TKIs or vehicle control. Mean ± SD are shown ($n = 9$ tumors for therapy naive, $n = 6$ tumors for each of the erALK cells). **d** Comparison of resistance levels of erALK-TKI cells evolved by gradual and acute drug exposure, measured by Cell Titer Glo assay. Mean ± SD of experimental triplicates are shown. **e** Sensitivity of independent derivates of erALK-TKI cells to the indicated ALK-TKIs, measured by crystal violet staining after 10 days of growing in the presence of indicated drugs. **f** Immunoblot analysis of the expression levels of the indicated proteins in the independent derivations of erALK-TKI cells, following 48 h growth in the absence of inhibitors to reduce their direct impact on cell signaling. "0" denotes gradually derived erALK-TKI cell lines, presented in **a**, "1–3" indicates independent derivate sub-lines obtained by acute selection for resistance in high drug concentrations, same as in **e**. Raw images shown in Supplementary Fig. 13. **g** UMAP analysis of single-cell RNA-seq expression of the indicated cell lines.

experimental system. This inference is consistent with time-lapse microscopy examination of the dynamics of drug resistance emergence, where robust growth in the presence of ALK-TKIs was observed after significant delay (Supplementary Video 1).

The existence of a distinct weakly resistant phenotypic state, which is a precursor to bona fide resistance, termed tolerance or persistence, has been widely studied in microbiology[17,18]. Based on the observation of a similar phenomenon in the context of response to TKIs, the term DTPs has been introduced to describe a weakly resistant subpopulation in EGFR + NSCLC[6] and later in other cancers[9,19,20]. Potentially, DTP cells can reflect a distinct pre-existing subpopulation or arise, either stochastically or deterministically, in response to drug-induced stress. To discriminate between these scenarios, we used tracing with selectively neutral DNA barcodes, an approach that has been previously used to demonstrate the pre-existence of EGFR-TKI resistance in EGFR + NSCLC[21]. Following the transduction of H3122 cells with a high-complexity lentiviral ClonTracer library at a low multiplicity of infection (MOI) (so that most of the transduced cells are labeled with a unique barcode) and elimination of non-transduced cells with puromycin selection, we achieved ~100× expansion of the barcoded cells. After taking a baseline aliquot, cells were split into parallel quadruplicate cultures and then exposed to 0.5 μM alectinib, lorlatinib, crizotinib, or DMSO control. After 4 weeks of incubation, barcode frequencies were enumerated by sequencing and were compared with the baseline frequencies (Fig. 2e). Evidence of both negative and positive selection was observed in all treatment groups (including DMSO controls), as barcode diversity, captured with Shannon diversity index, decreased (Supplementary Fig. 3a), whereas several subpopulations expanded (Supplementary Fig. 3b). Spearman's ranking of positively selected barcodes revealed a strikingly high degree of correlation between replicates, indicating pre-existence of stable weakly resistant subpopulations (Fig. 2f). However, correlation between samples treated with different ALK-TKIs was either absent or much less pronounced, indicating that distinct selective pressures exerted by different ALK-TKIs might amplify distinct pre-existing tolerant subpopulations. Unsupervised hierarchical clustering analysis revealed a partial overlap between positively selected subpopulations across multiple ALK-TKIs, indicating that some of the pre-existent phenotypes were fit under multiple ALK-TKIs (Fig. 2g). In contrast to the fitness cost of classical persistence under baseline growth conditions, considered to be a form of bet hedging[18],

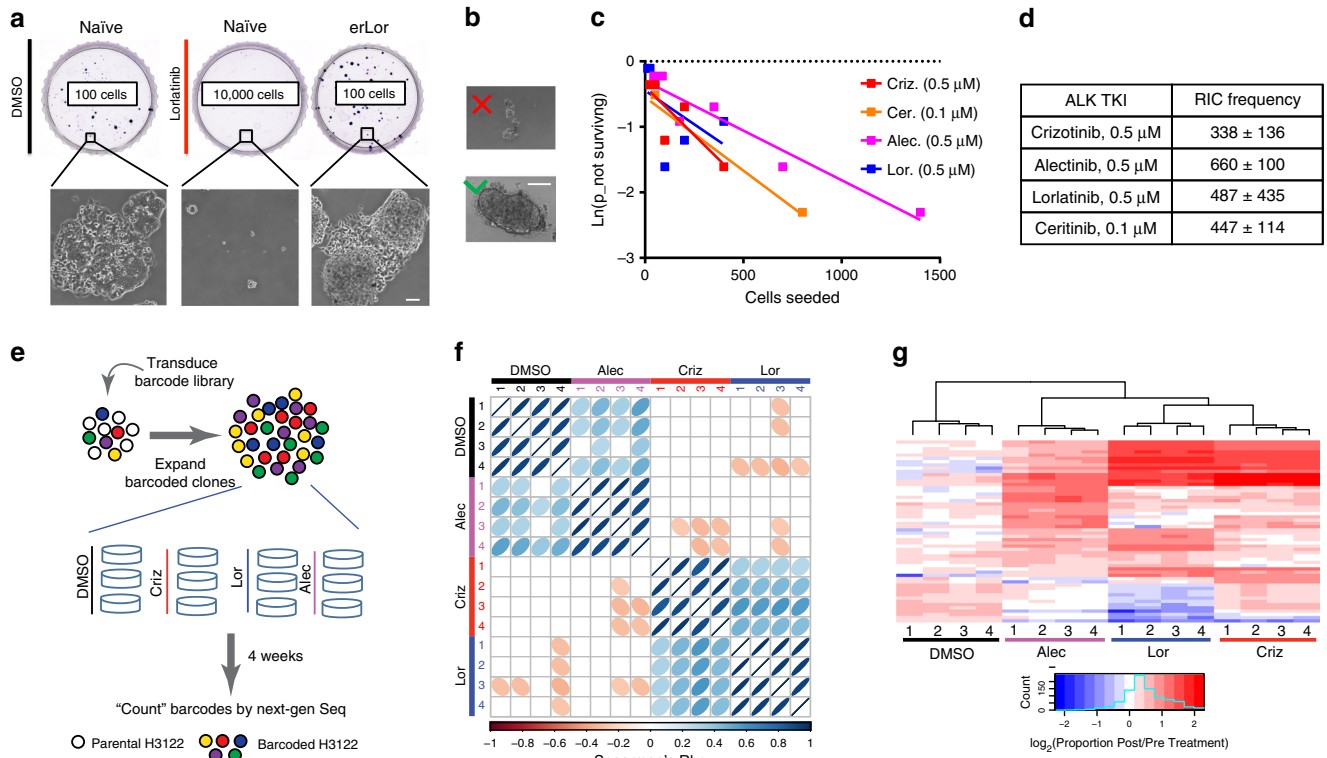

**Fig. 2 Pre-existence of diverse ALK-TKI tolerant subpopulations. a** Representative images of crystal violet stained whole plates and microscopic images of colonies (at ×10 magnification) following 10 days of culture in DMSO or 0.5 μM lorlatinib. **b** Illustration of the size cut-off criteria used for quantification of limiting dilution experiments. The scale bars in **a** and **b** represent 100 μm. **c** Fitting of limiting dilution assay data and **d** quantification of the frequency of resistance-initiating cells. **e** Schemata of the clone-tracing experiment. **f** Spearman's pairwise correlation analysis of positively selected barcodes, which have reached frequency, exceeding the highest barcode frequency in the baseline aliquot sample, between samples indicated in X and Y axes. Numbers indicate biological replicates (separate dishes). **g** Hierarchical clustering analysis of the positively selected barcodes shown in **f**. Columns indicate individual biological replicates. Rows indicate individual barcodes.

subpopulations enriched in ALK-TKIs, on average, were also slightly enriched under the DMSO control, indicating the lack of a baseline fitness penalty (Supplementary Fig. 3c). Thus, taken together, this reveals that resistance to an ALK-TKI in H3122 cells originates from pre-existent, heterogeneous subpopulations with ALK-TKI-specific pan-TKI tolerance.

**Gradual development of ALK-TKI resistance**. Next, we asked how tumor cells progress from tolerance toward full resistance. The prevalent assumption in the modeling, experimental, and clinical communities is that resistance results from a single-hit transition via acquisition of a resistance-conferring mutation or an epigenetic switch[2,22]. Should this be the case, a binary distribution of growth rates, corresponding to tolerant and resistant cells, would be expected in clonogenic assays after ALK-TKI exposure, with longer exposure times leading to a higher proportion of large resistant colonies (Fig. 3a). As expected from the predicted elimination of sensitive subpopulations, clonogenic proportion in 0.5 μM crizotinib progressively increased from the initial 2.6% to 26% at week 3, whereas the clonogenic proportion in 0.5 μM lorlatinib increased from 1% to 17% (Fig. 3b). However, although longer ALK-TKI exposure lead to an increase in the average colony size, this increase was apparently homogeneous, suggesting a gradual development of resistance (Fig. 3c). Analysis of the colony size distributions at the intermediate (2 and 3 weeks) time points using Kolmogorov–Smirnov (KS) statistics revealed that the observed colony sizes cannot be explained by mixed sampling from distributions of tolerant and erALK-TKI

cells (Supplementary Fig. 4a, b). Consistent with the lack of tolerance-associated proliferation penalty in the barcoding experiment inferences, we did not observe substantial differences in the size of colonies formed by cells pre-treated by crizotinib or lorlatinib for varying durations of time in the absence of the drugs (Supplementary Fig. 4c, d).

KS analyses consider transitions from tolerance to resistance within the 1–3 weeks of growth under ALK-TKI prior to the clonogenic assay. However, the transition could also occur during colony growth. Therefore, to test compatibility of the experimental data with a single-step transition more rigorously, we developed an agent-based mathematical model of the experimental assay, which simulated growth both prior and during the clonogenic assay across a range of (epi)mutational probabilities and numbers of (epi)mutational steps (Fig. 3d, e and Supplementary Methods). To account for the variability in observed colony sizes, the maximal proliferation probability was set for each simulated colony individually, based on random sampling of sizes of colonies produced by resistant cells. We calibrated the initial proliferation probability using just the median colony size, assuming minor variability in the proliferation rates of such cells. At each cell division, a cell can increase in division rate, reflecting adaptive (epi-)mutations. Cells can transition from the initial to maximal division probability by either a single (epi)mutational step ($n = 1$) or multiple ($n > 1$) steps, representing fractional increments of the single transition (Fig. 3d and Supplementary Methods).

Whereas the initial and maximal growth rates are fixed based on the experimental data, the intermediate rates within in silico

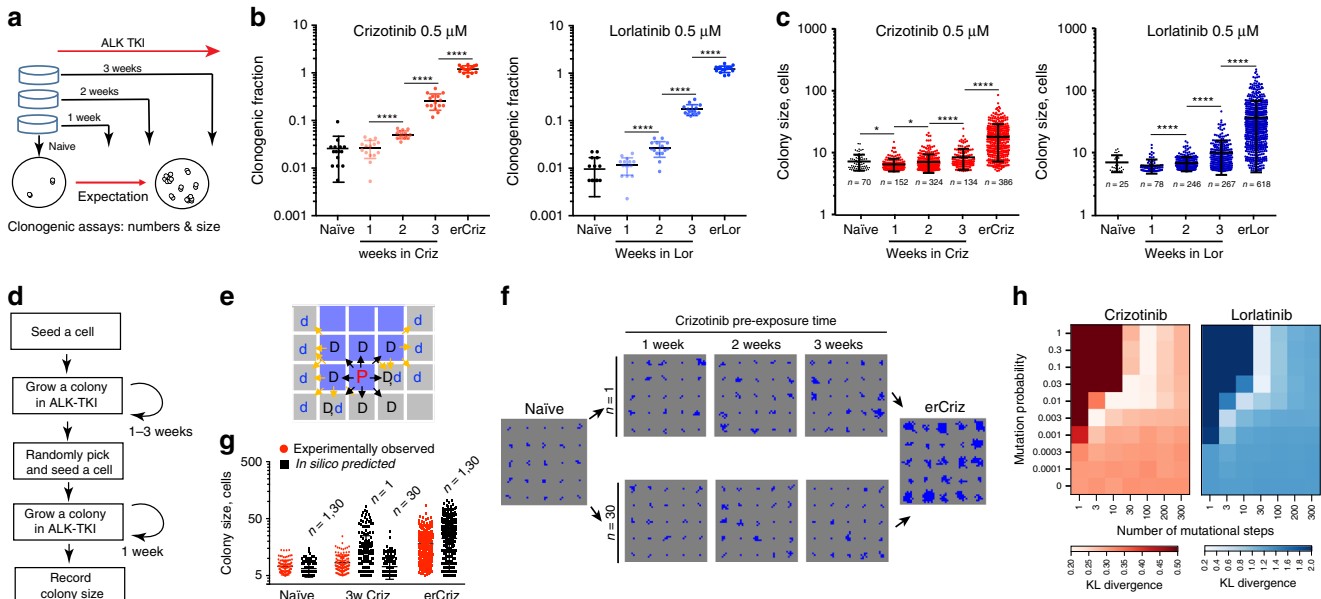

**Fig. 3 Graduality of evolution of ALK-TKI resistance. a** Experiment schemata. Therapy-naive cells were pre-cultured in the presence of crizotinib or lorlatinib for 0–3 weeks, then seeded at clonogenic densities in the presence of ALK-TKIs or DMSO control. After 7 days, numbers and sizes of colonies are determined. **b** Clonogenic survival in the presence of the indicated ALK-TKIs after pre-incubation for indicated times; data are normalized to clonogenic survival in DMSO control. Mean ± SD of 15 replicates (separate wells) is shown. **c** Distributions of colony sizes in the indicated ALK-TKI. *$p < 0.05$ and **$p < 0.0001$ of a Mann–Whitney test. Mean ± SD of individual colonies are shown. **d** Logical flow diagram for the agent-based model, simulating growth during both pre-incubation and the clonogenic assay. **e** Proliferation space check scheme. Cells are seeded into a 2d lattice that simulates the surface of a culture dish. If space is available (no more than one cell separating the cell from an empty space), a cell can proliferate with a given probability inferred from the experimental data. Blue and gray cells denote occupied and empty spaces respectively. "P" stands for the parent cell, "D" for daughter cell, "d" for displaced cell. Black arrows indicate options for placement of daughter cells, yellow arrows indicate options for displaced cells. Proliferation can occur if a nearby space is either immediately available or separated by a single cell, in which case this cell is pushed into an empty space, with an extra copy of the proliferating cell displacing it. **f** Example of colony growth simulations, initiated from cells pre-incubated in crizotinib for the indicated time, during the clonogenic growth phase off the assay, contrasting $n = 1$ vs. $n = 30$. **g** Comparing divergence between in silico and experimental data, with $n = 1$ and $n = 30$ (epi)mutational steps. Mean ± SD of individual experimental and simulated colonies are shown. **h** Kullback–Leibler divergence-based comparison of the experimental data with the outcomes of simulations, covering parameter space for the indicated mutation probabilities and numbers of mutational steps.

can differ based on two parameters: (epi)mutational probability and the number of mutational steps required to reach maximum proliferation rate. We explored outcomes with the full range of possible (epi)mutational probabilities (0–1) and the numbers of mutational steps ranging from 1 to 300. With each choice of parameters, we generated 100,000 in-silico stochastic simulations (Fig. 3f and Supplementary Video 2). We found that a single mutational step provided the poorest fit to the data for all mutation probabilities. The best fit was achieved with a number of mutational steps in the range of 3–100 (Fig. 3g, h). Considerations of cell death, bi-directionality of (epi)mutational changes in cell fitness and variability in fitness effects of (epi) mutations did not change the poor fit of a single-step transition (Supplementary Fig. 4e, f and Supplementary Methods). Therefore, our analyses are inconsistent with the prevalent assumption of resistance achieved through a single (epi)mutational change, which converts tolerance to resistance, and instead support a gradual improvement of cell fitness under therapy.

Next, we decided to characterize the mechanistic underpinning of the evolution of resistance. Extensive prior studies of mechanisms of resistance to ALK-TKIs, including studies using the H3122 model enabled us to ask whether the previously identified ALK-TKI resistance mechanisms are consistent with a single-hit acquisition of resistance. Clinical resistance to ALK inhibitors is frequently associated with point mutations in the kinase domain of ALK, which reduce drug binding[23]. However, targeted Sanger sequencing of PCR-amplified cDNA revealed lack of hotspot ALK mutations in erALK-TKI cell lines

(Supplementary Fig. 5). Given the frequent association of clinical ALK-TKI resistance with *EML4-ALK* amplification[24] and the observed increase in the expression of EML4-ALK in some of the erALK-TKI-resistant cell lines (Fig. 1f), we interrogated EML4-ALK amplification status in the treatment-naive and erALK-TKI-resistant cells (lines "0" from Fig. 1f), using the mutational break-apart fluorescence in-situ hybridization assay. The majority of treatment-naive H3122 cells displayed four copies of the wild-type allele and one copy of the fusion allele, with a minor subpopulation where the fusion gene signal could not be detected. Some of the erALK-TKI cells displayed amplification of the mutant allele (Fig. 4a). Extrachromosomal amplification of oncogene-containing DNA has been recently implicated in the rapid evolution of TKI resistance[25]; however, examination of metaphase spreads revealed that the amplified alleles were localized within the same chromosome. Notably, we observed substantial heterogeneity in the amplification status of *EML4-ALK*, both between and within erALK-TKI cell lines (Fig. 4b). The majority of erCriz cells and a fraction of erAlec and erCer contained amplified *EML4-ALK*. In contrast, erLor cells not only lacked *EML4-ALK* amplification but also contained a significantly higher proportion of cells with undetectable mutant allele ($p < 0.0001$ in a $\chi^2$-test) compared with the treatment-naive cells, suggesting that genomic loss *of EML4-ALK* might be selectively advantageous under the more potent ALK-TKI.

To investigate the functional importance of the observed changes in *EML4-ALK* copy numbers, we transfected treatment-naive erCriz and erLor cells with constructs co-expressing Cas9

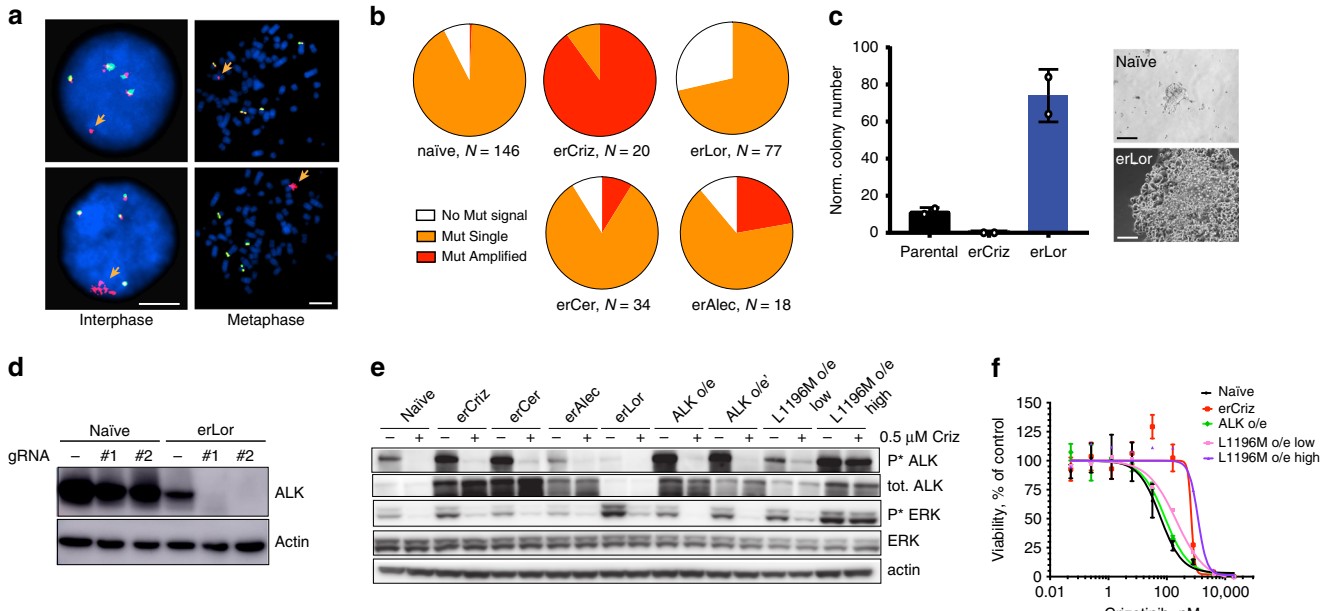

**Fig. 4 Impact of ALK mutation and amplification on TKI sensitivity. a** Representative images for interphase and metaphase FISH analysis for EML4-ALK fusion and amplification status. Separation of 3′ (red) probe from 5′ (green) probe indicates ALK fusion event (orange arrows). The scale bars represent 5 μm. **b** Frequency of cells with the indicated EML4-ALK fusion and amplification status in the gradually evolved erALK-TKI cell lines (lines 0 were analyzed). **c** Impact of CRISPR-mediated genetic ablation of ALK on clonogenic survival of the indicated H3122 derivates. Mean ± SD of experimental duplicates, representing separate dishes with alternative ALK directed guide RNAs; representative colonies are shown. The scale bars represent 100 μm. **d** Evaluation of EML-ALK ablation by immunoblotting analysis. Raw images shown in Supplementary Fig. 14. **e** Immunoblot evaluation of the expression and activity of EML4-ALK oncogenic signaling in the presence of Crizotinib or after 48 h of drug holidays, for the indicated cell lines with evolved and engineered resistance. ALK o/e and ALK o/e' denote independently derived sublines. Raw images shown in Supplementary Fig. 15. **f** Impact of retrovirally mediated overexpression of EML4-ALK fusion and its L1196M mutant variant on sensitivity to crizotinib, measured by Cell Titer Glo assay. Mean ± SD of experimental triplicates representing separate wells are shown.

and one of two different ALK-targeting guide RNAs, and selected for puromycin-resistant colonies. No colonies could be observed for erCriz cells, suggesting a critical dependency on EML4-ALK (Fig. 4c). Naive H3122 cells formed few small colonies, resembling tolerant colonies formed upon exposure to an ALK-TKI (Fig. 2a). Puromycin-resistant naive cells, transfected with guide RNA directed against ALK expressed EML4-ALK protein, displayed normal ALK expression (Fig. 4d), indicating a strong selective disadvantage of losing EML4-ALK expression and selection of variants that uncouple antibiotic resistance from guide RNA expression. In contrast, erLor cells formed multiple large colonies consistent with a lack of growth inhibition (Fig. 4c), despite complete ablation of the protein expression of the *EML4-ALK* gene (Fig. 4d). This observation is consistent with reduced baseline EML4-ALK expression in erLor cells (Fig. 1f) and suggests that erLor cells completely lose EML4-ALK addiction.

Given that EML4-ALK amplification resulting in overexpression is considered to provide a bona fide resistance mechanism to ALK inhibition[26], we asked whether the observed increase in EML4-ALK expression is sufficient to account for ALK-TKI resistance. To this end, we retrovirally overexpressed EML4-ALK protein in H3122 cells, resulting in levels of total and phosphorylated EML4-ALK, which closely resemble those observed in EML4-ALK-amplified erALK-TKI cells (Fig. 4e). After exposure to crizotinib, these cells retained residual levels of ALK phosphorylation similar to those observed in the erALK-TKI cells (Supplementary Fig. 6a). However, cells with EML4-ALK overexpression displayed only a marginal increase in crizotinib resistance (Fig. 4f), suggesting that although ALK amplification contributes to resistance, it is insufficient to fully account for it.

Given the insufficiency of EML4-ALK overexpression to confer full resistance, we interrogated the functional impact of the most common resistance-associated point mutation, L1196M, which is considered to be a clear case of a single hit resistance mechanism to crizotinib[26]. Surprisingly, at low expression levels, achieved with <1 retroviral MOI, L1196M expression only moderately decreased crizotinib sensitivity (Fig. 4f). Higher overexpression levels of the mutant protein blocked crizotinib from downregulating phosphorylation of EML4-ALK, as well as its main downstream effector ERK (Fig. 4e and Supplementary Fig. 6a, b), and provided crizotinib resistance at a level that is similar or higher to that observed in erCriz cells (Fig. 4f). Whereas the results with L1196M overexpression are consistent with previously reported sufficiency to confer resistance in NIH-H3122 cells[27,28], and ALK mutations were reported co-occur with EML4-ALK amplification in some patients[29,30], these observations suggest that two mutational hits (mutation and overexpression) might be required to achieve full resistance. Thus, common resistance-associated mutational changes, assumed to provide single hit resistance, reduce drug sensitivity but do not provide for a full adaptation to the ALK-TKI induced selective pressures.

Examination of resistant cell lines with Oncomine Focus Assay[31] failed to detect additional common mutations. However, CytoScanHD single-nucleotide polymorphism (SNP) array revealed additional genetic changes, including recurrent chromosomal amplifications in chromosomes 2, 3, 12, and 17. Interestingly, two out of three examined erLor lines displayed Chr12 p12.1-p11.1 amplification containing *KRAS*, whereas a third one contained Chr1 p13.2-p12 amplification containing *NRAS* (Supplementary Fig. 7a). Notably, chromosomal amplification of genomic regions containing *KRAS* and *NRAS* were

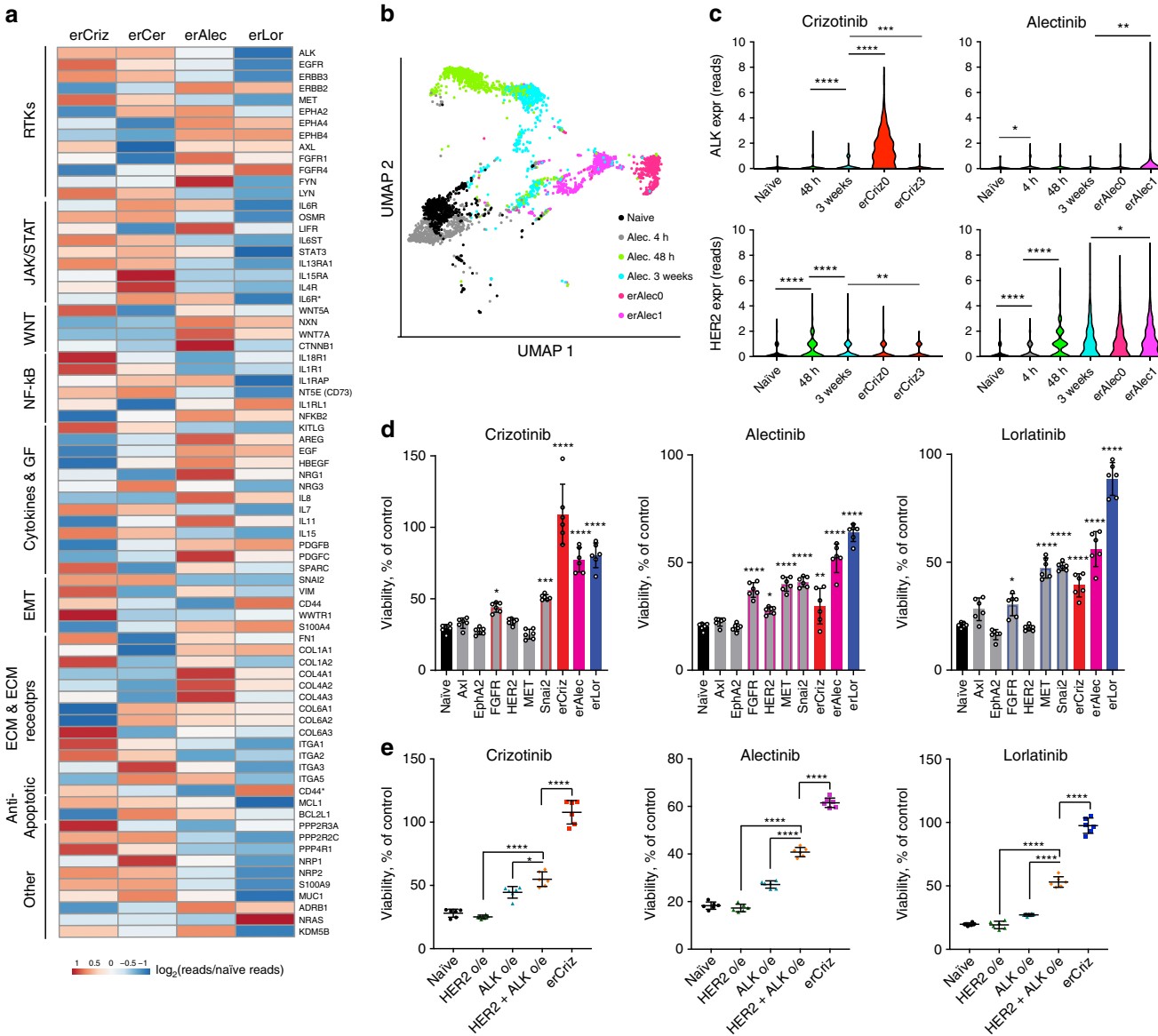

**Fig. 5 ALK-TKI resistance integrates multiple mechanisms. a** Normalized mRNA expression for the indicated genes, previously associated with chemotherapy or targeted therapy resistance, in erALK-TKI cell lines across the indicated functional categories. **b** UMAP analysis of single-cell mRNA expression data from cells exposed to 0.5 μM alectinib for the indicated time duration. **c** Violin plot of expression levels of ALK and HER2 following indicated duration of exposure to the indicated ALK-TKIs. *$p < 0.05$, **$p < 0.01$, ***$p < 0.001$, and ****$p < 0.0001$ of a Mann–Whitney $U$-test. **d** ALK-TKI sensitivity of engineered cell lines, lentivirally overexpressing indicated genes to the indicated ALK-TKIs (0.5 μM), as determined by Cell Titer Glo assay. Mean ± SD of experimental replicates ($n = 6$, representing separate wells) are shown. **e** Impact of combination of individual resistance mechanisms toward sensitivity to indicated ALK-TKIs (0.5 μM), determined by Cell Titer Glo assay. *$p < 0.05$, **$p < 0.01$, ***$p < 0.001$, and ****$p < 0.0001$ of ANOVA analysis with Dunnett's (**d**) or Tukey's (**e**) multiple comparison correction. Mean ± SD of experimental replicates ($n = 6$, representing separate wells) are shown.

associated with elevated transcript levels of these proto-oncogenes (Supplementary Fig. 7b).

Given the growing evidence for the importance of non-genetic mechanisms in therapy resistance, we used RNA sequencing (RNA-seq) analysis to examine changes in gene expression, accompanying erALK-TKI phenotypes. After a 48 h drug holiday (to reduce the direct impact of ALK-TKI on gene expression), erALK-TKI cells displayed multiple gene expression changes, previously implicated in TKI and chemotherapy resistance, including increased expression of multiple receptor tyrosine kinases (EGFR, HER2, FGFR, AXL, EPHA2, etc.), cytokines, extracellular matrix (ECM) and ECM receptors, and other types of molecules, suggesting a complex, multifactorial nature of resistant phenotypes (Fig. 5a). Co-expression analysis of cell lines

in the CCLE database revealed that the resistance-associated genes, upregulated in erALK-TKI cells, belonged to distinct gene co-expression clusters, suggesting that the resistant phenotypes cannot be explained by a single coordinated transcriptional switch (Supplementary Fig. 8a, b), whereas gene set enrichment analysis revealed several shared enriched gene sets (Supplementary Fig. 8c). Interestingly, all of the examined resistant cell lines displayed an epithelial to mesenchymal transition (EMT) signature. Whereas EMT has been described as one ALK-TKI resistance mechanism[32,33], our results suggest a need for a more nuanced interpretation, given the predictably distinct phenotypic characteristics of erALK-TKI cell lines, evolved under different ALK-TKIs, as these differences would be missed under the umbrella of the EMT/stemness explanation.

To gain further insights into the dynamics of phenotypic changes during the evolution of resistance, we analyzed single-cell transcriptomes after different exposure times to ALK-TKIs. UMAP analysis revealed gradual phenotypic progression from naive to resistant phenotypes (Fig. 5b and Supplementary Fig. 9). Even a brief (4 h) alectinib exposure substantially impacted cell phenotypes, suggesting that acquisition of resistance might reflect not only the action of drug-imposed selection but also direct drug-induced cell adaptation. To gain further insight into the temporal dynamics of the acquisition of resistance-conferring expression changes, we analyzed the expression of ALK and HER2 at single-cell levels, at different time points, after exposure to an ALK-TKI. Consistent with high levels of ALK expression and genetic amplification of EML4-ALK in erCriz cell lines, the expression of ALK increased upon exposure to crizotinib, but not alectinib. In contrast, expression of HER2 became elevated within 48 h of exposure to both drugs, suggesting a direct adaptive response, then decreased upon prolonged incubation with crizotinib, while staying upregulated or increasing further under alectinib (Fig. 5c).

To explore possible epigenetic mechanisms underlying the observed stable changes in gene expression in erALK-TKIs cell lines, we analyzed the global repatterning of Histone 3.3 lysine 27 acetylation (H3K27ac), a posttranslational histone modification associated with strong enhancer elements using chromatin immunoprecipitation sequencing (ChIP-Seq) analysis. We found distinct global patterns of H3K27 acetylation at gene regulatory elements between naive and ALK-TKI-resistant lines (Supplementary Fig. 10a). Notably, statistically significant changes in H3K27 acetylation were observed in regulatory elements of genes with resistance-associated gene expression changes (Supplementary Fig. 10b), including genes previously implicated in therapy resistance. For example, consistent with observed differences in protein expression (Fig. 1f), we found new H3K27ac peaks in the vicinity of the ERBB2/HER2 gene in erAlec and erLor lines, but not in the erCriz lines (Supplementary Fig. 10c). New peaks were also observed near genes with increased genomic copy numbers (EML4, K-RAS, and N-RAS) (Supplementary Fig. 10d), suggesting that that stable upregulation of resistance-associated genes might result from a combined input of genetic and epigenetic changes.

To evaluate the functional significance of these transcriptional changes, we tested the impact of overexpression of selected genes previously implicated in TKI resistance. Lentiviral overexpression of HER2, FGFR, AXL, and SLUG significantly increased resistance to multiple ALK-TKIs (Fig. 5d). However, resistance levels observed in these engineered cell lines fell short of the levels observed in the erALK-TKI lines, suggesting insufficiency of individual mechanisms to fully account for resistance. On the other hand, combined overexpression of EML4-ALK and HER2 led to significantly higher resistance levels, compared with cells overexpressing either EML4-ALK or HER2 separately, although still failing to recapitulate resistance levels observed in erALK-TKI cells (Fig. 5e). These results support the notion that ALK-TKI resistance reflects a combined output of multiple genetic and epigenetic changes, which contribute to increased fitness in an additive or synergistic manner.

**Temporal collateral sensitivities of resistance-evolving cells**. It is frequently assumed that, according to the principle of an evolutionary fitness tradeoff[34], resistance-conferring phenotypes are associated with strong fitness penalties outside of the drug[35–38]. This fitness penalty could enable evolutionary-informed adaptive therapy, by creating a tug of war between therapy sensitive and therapy-resistant populations with strategic treatment breaks[37]. Thus, we examined growth rates of erALK-TKI cell lines in the

presence and absence of the drugs. Whereas in some cases cells taken off the drugs indeed proliferated slower than drug-naive control, some of the resistant cell lines proliferated at similar or even higher rates (Supplementary Fig. 11a). Interestingly, all of the examined erCriz and one out of three of the examined erAlec cell lines displayed higher rates of proliferation in the presence of the inhibitors, consistent with previously reported observations in melanoma[39,40]. The remaining cell lines were either modestly inhibited, or unaffected by the ALK-TKI used for their selection (Supplementary Fig. 11a).

Fitness of tumor cells is context dependent[41,42] and in vitro two-dimensional cultures do not capture death/proliferation dynamics within tumors in vivo. Further, non-cell autonomous interactions between phenotypically distinct subpopulations could significantly alter fitness in competitive settings[43,44]. Therefore, we compared baseline fitness of differentially labeled (green fluorescent protein (GFP) and mCherry) naive and erALK-TKIs cells in a competitive in-vivo context by implanting their mixtures either subcutaneously or into the lungs (via tail vein injections). Only erCriz cells in the lungs displayed signs of reduced fitness compared with therapy-naive cells (Supplementary Fig. 11b). In contrast, erLor cells had a significant selective advantage over naive cells both in subcutaneous and lung tumors. Therefore, resistance is not necessarily associated with a fitness penalty and might be linked to higher fitness outside of the drugs.

On the other hand, growth out of the primary therapy is just one of many potential contexts where an evolutionary tradeoff might be manifested. Application of a different drug, to which ALK-TKI-resistant phenotypes are collaterally sensitive, could provide an alternative strategy to create a fitness tradeoff[45,46]. Given the frequent upregulation of EGFR and HER2 in cells, resistant to the front-line ALK-TKI alectinib, we tested whether the dual EGFR/HER2 inhibitor lapatinib could create a collateral fitness tradeoff. Whereas H3122 cells failed to develop resistance to alectinib in the presence of lapatinib (Supplementary Fig. 12a), erAlec cells were not sensitized to lapatinib as a single agent and only some of the independently derived erAlec cell lines were inhibited by the combination of alectinib and lapatinib (Supplementary Fig. 12b). At the same time, evolving cells were remarkably sensitive to lapatinib, even as a single agent, although this sensitivity gradually diminished as cells became more resistant to alectinib (Fig. 6a, b). This collateral sensitivity cannot be explained by the pre-existence of lapatinib-sensitive, alectinib-tolerant subpopulations, as pretreatment with lapatinib for up to 3 weeks did not substantially impact sensitivity of H3122 to alectinib (Supplementary Fig. 12c). Consistently, administration of lapatinib as a single agent, after 3 weeks in alectinib, or as a combination therapy, led to complete elimination of H3122 cells in vitro (Fig. 6c).

Encouraged by this observation, we asked whether alectinib/ lapatinib cycling or a combination treatment with the two drugs could outperform alectinib monotherapy in vivo. We found that lapatinib was completely ineffective in adaptive cycling with alectinib, as tumor regrowth during lapatinib cycles was indistinguishable from the regrowth observed with a drug holiday (Fig. 6d). Most likely, this lack of efficiency in vivo reflects its inability to reach the high concentrations required to achieve collateral sensitivity in vitro. Still, lapatinib significantly increased tumor sensitivity to alectinib in a combination therapy setting (Fig. 6d). Whereas, at this point, the clinical utility of our observations remains unclear, they provide a proof of principle that temporally restricted collateral vulnerabilities of evolutionary intermediates might be exploited therapeutically to improve responses, and potentially, to forestall the emergence of resistance in targetable lung cancers.

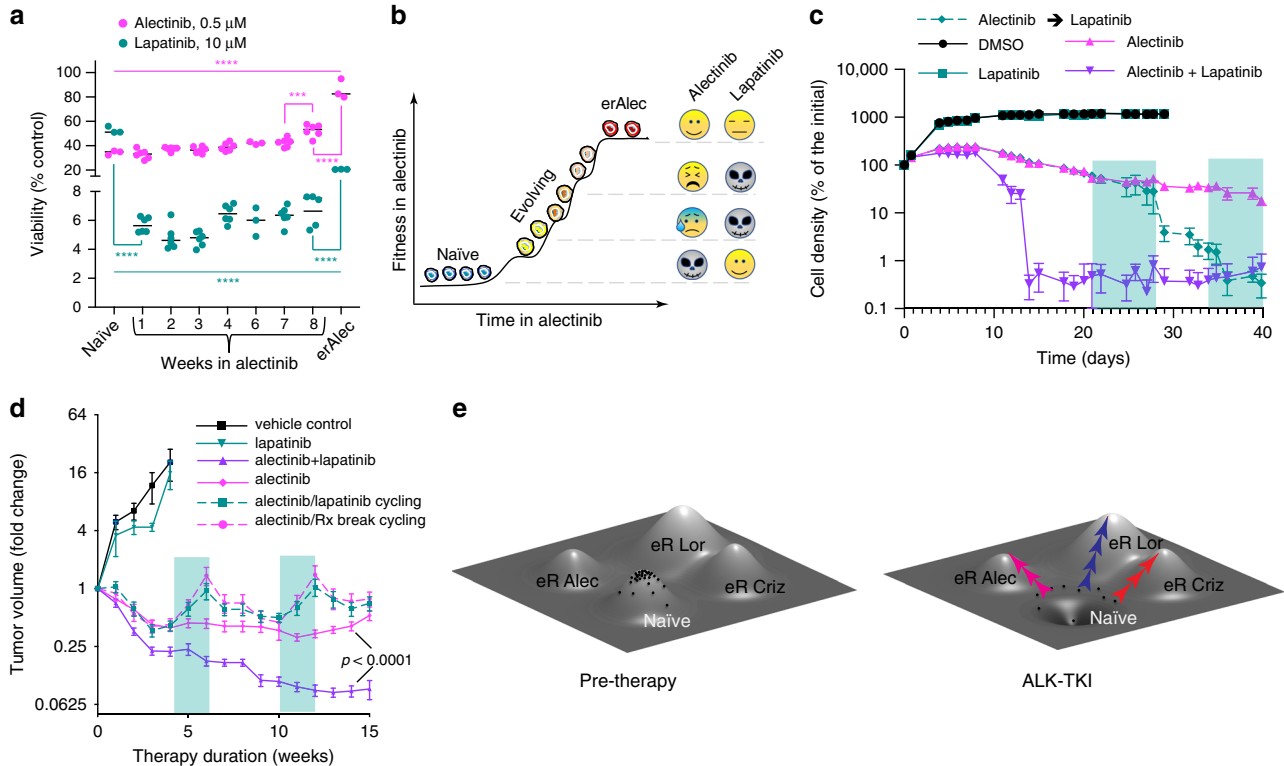

**Fig. 6 Collateral sensitivities of evolutionary intermediates. a** Impact of pre-exposure to 0.5 μM alectinib for the indicated periods on sensitivity to alectinib and lapatinib, measured by Cell Titer Glo assay. One-way ANOVA was used for both drugs ($p < 0.0001$). Adjacent time points were compared using Sidak's multiple comparison tests (***$p < 0.001$ and ****$p < 0.0001$). Mean ± SD of experimental replicates are shown. $n = 3$, representing separate wells, for naive control and erAlec H3122 cells. $n = 6$ wells (3 wells each of 2 biological replicates) for cells pre-exposed to alectinib. Data for one of the biological replicates is missing in week 6. **b** Illustration of association of evolving resistance to alectinib with collateral sensitivity to lapatinib. **c** Lapatinib can prevent the development of resistance to alectinib in vitro, both as a combination treatment, or in drug cycling. Green shading indicates switching from alectinib to lapatinib monotherapy. Residual signal in the combination therapy and cycling groups reflects autofluorescence, as visual examination revealed lack of surviving tumor cells. Mean ± SD of experimental replicates ($n = 3$ representing separate wells for DMSO and lapatinib monotherapy, $n = 6$ for the remaining groups) are shown. **d** Change in volume of H3122 xenograft tumors treated with indicated therapies; treatment was initiated 3 weeks post tumor implantation. Green shading indicates switching from alectinib to lapatinib monotherapy or vehicle control. Mean values ± SE are shown; $n = 6$ for lapatinib monotherapy, $n = 10$ for alectinib monotherapy, switching between alectinib and lapatinib, and alectinib/lapatinib combination, $n = 9$ for vehicle control, replicates represent separate tumors. **e** Evolving resistance interpreted through a fitness landscape metaphor. Naive cells occupy a local fitness peak. Drug exposure reshapes the landscape, turning this fitness peak into a fitness trough. Different ALK inhibitors act on partially distinct outliers, directing their evolution toward distinct fitness peaks.

## Discussion

Despite substantial advances in deciphering the molecular mechanisms of resistance to TKI-based targeted therapies and the development of more effective drugs, advanced targetable lung cancers remain incurable, as tumors eventually acquire resistance and relapse. Developing strategies to interfere with evolving resistance has the potential to substantially improve long-term survival outcomes. However, this is contingent on a correct understanding of the underlying evolutionary dynamics. In contrast to the large body of experimentally derived knowledge on individual molecular mechanisms of resistance, our understanding of its evolutionary causes and dynamics remains less developed. The subject of evolutionary-informed therapies has received significant attention from mathematical modelers[5,16,47,48]. However, given the paucity of experimental data acquired to derive solid assumptions for building models, these modeling studies often have to rely on conjectures from reductionist, mechanism-centered studies, thus limiting their potential.

Acquired resistance is commonly viewed through two major conceptual frameworks. According to the first, currently dominant gene-centric framework, resistance arises from a selective expansion of genetically or epigenetically distinct (meta)stable subpopulations. The subpopulation(s) could either pre-exist therapy or arise de novo from mutational conversion of sensitive or tolerant cells. In both cases, clinical and experimental studies operating within this paradigm typically reduce acquired resistance to a single cause, such as a point mutation (e.g., the L1196M gatekeeper mutation)[30], gene amplification (e.g., EML4-ALK[30], cMET[49], etc.), or a non-mutational stable change in the expression of a resistance-conferring gene (e.g., IGFR)[50]. Consequently, the assumption of a single hit resistance mechanism is highly prevalent in the mathematical modeling community[5,16,22].

The second framework views resistance as the result of drug-induced reprogramming, where phenotypic plasticity enables tumor cells to directionally rewire their signaling, metabolic, and gene-expression networks to cope with inhibitor-induced perturbations[9,51,52]. In this system-biology-based framework, resistance can be gradual and multifactorial; however, the causal role of selection is sometimes rejected due to its link with mutation-centric paradigms[9].

The two frameworks are not strictly mutually exclusive, as they can be bridged in a two-step process involving reprogramming-mediated formation of tolerant subpopulations followed by an

(epi)mutational switch to full resistance. Our results, however, suggest an alternative scenario of gradual, multifactorial resistance, which integrates features of the two paradigms described above. We speculate that selective pressures imposed by therapies act on phenotypic heterogeneity, stemming from both stochastic and drug-induced changes, leading to a gradual increase in population fitness through acquisition of additional genetic and epigenetic changes until a local fitness peak is reached (Fig. 6e). Different evolutionary trajectories under distinct ALK-TKIs reflect inhibitor-specific differences in both selective pressures and direct drug-induced adaptive responses, which can become hardwired if selectively advantageous. This model is closer to Darwin's original thesis "As natural selection acts solely by accumulating slight, successive, favorable variations, it can produce no great or sudden modifications"[53] rather than the currently prevalent mutation-centric re-interpretation of Darwinian selection within the clinical and basic science cancer research community[54].

Gradual, multifactorial acquisition of resistance has been recently observed in an elegant experimental study of acquired BRAFi resistance in melanoma[9]. The authors rejected the Darwinian explanation based on the lack of pre-existent resistance and thus interpreted their observations strictly within a reprogramming paradigm. We argue that their data are highly consistent with our model, which does not reduce Darwinian selection to genetically distinct pre-existing variants. Likewise, a recent single-cell sequencing study in triple-negative breast cancer demonstrated that although resistance to chemotherapy can be traced to pre-existent genetically distinct subpopulations, resistant cells acquire new phenotypic changes, indicating epigenetic modlifications[55]. Similarly, a recent study in hormone positive breast cancers demonstrated a multi-step underpinning of resistance, involving both genetic and non-genetic changes[51]. Therefore, our inferences with a model of ALK mutant NSCLC might be generalizable to a broader range of scenarios of acquired therapy resistance.

Finally, our data suggests that evolutionary intermediates of acquired resistance may present temporally restricted opportunities for therapeutic interventions, which is consistent with the previously reported existence of collateral sensitivities in mutational intermediates in the evolution of resistance in Ph+ ALL[56]. These findings support the notion that explicit interrogation and nuanced understanding of the evolutionary mechanisms of acquired drug resistance could open the way to designing evolutionary-informed therapy approaches, focused on staying a step ahead of resistance, rather than just reacting to it when it (inevitably) arises.

## Methods

**Derivation of resistant cell lines**. Parental and ALK-TKI-resistant H3122 cells were grown in RPMI (Gibco, ThermoFisher) supplemented with 10% fetal bovine serum (FBS) (Serum Source), penicillin/streptomycin, insulin (Gibco), and anti-clumping agent (Gibco). ALK-TKI-resistant cell lines were generated by further evolving lines used in ref. [10], derived through a progressive increase in ALK-TKI concentrations; eventually, they were maintained in 0.5 μM crizotinib, 0.2 μM ceritinib, 2.0 μM alectinib, and 2.0 μM lorlatinib. Alternatively, for the acute derivation, resistant lines were derived by exposing treatment-naive H3122 cells directly to the high ALK-TKI concentrations (0.5 μM crizotinib, 2 μM alectinib and lorlatinib, or 100 nM ceritinib), re-plating every 2–4 weeks to relieve spatial growth constraints over the course of 2–4 months.

**Short-term viability assays**. Short-term cell viability was measured using Cell Titer Glo reagent (Promega). Typically, 2000 cells per well were plated in a 96-well plate (Costar). Drugs were added 24 h later and the assays were performed using manufacturer-recommended protocol 3–4 days after drug addition. Viability was calculated first by subtracting the luminescence of empty wells then by division by average DMSO luminescence. Dose–response curves were generated by fitting the following equation to the data: $y = b + \frac{100 - b}{1 + \left(\frac{x}{IC_{50}}\right)^k}$, where $y$ is luminescence, $x$ is drug

concentration, $IC_{50}$ is the half maximal inhibitory concentration, $k$ is the hill slope, and $b$ is the luminescence as the drug concentration approaches ∞.

**Clonogenic assays**. Cells were plated in the presence of ALK-TKIs or DMSO vehicle control at varying densities into 6 cm dishes or multi-well plates in duplicates or triplicates, and were grown for 10 days, at which point they were fixed and stained with crystal violet, following protocols described in ref. [57]. To measure the evolution of gradually increasing resistance, nuclear mCherry expressing H3122 cells were plated at ~400,000 cells per 6 cm dish, allowed to attach overnight, and exposed to DMSO vehicle control, 0.5 μM crizotinib, or 0.5 μM lorlatinib the following day. After culturing for 1–3 weeks, cells were collected and seeded in 96-well plates (Costar) at 50 cells/well for DMSO control, or between 50 and 500 cells for crizotinib and lorlatinib. Number of colonies and colony sizes were measured 1 week later for colonies larger than 1000 pixels (~5 cells), based on fluorescent area. To minimize the impact of variability in seeding numbers, clonogenic survival in the presence of ALK inhibitors was normalized to clonogenic data in the DMSO controls.

**Determining frequency of resistance-initiating cells**. Cells were plated with initial seeding densities of 400, 400, 800, 1400, and 400 cells/well in DMSO (0.1%), crizotinib (0.5 μM), ceritinib (0.1 μM), alectinib (0.5 μM), and lorlatinib (0.5 μM) treated plates, respectively, in a 96-well plate (Falcon). Five additional 2× dilutions were generated from these wells, each with 10 separate wells. ALK inhibitors were added after 24 h. After 7 weeks of treatment, wells containing no colonies larger than ~50 cells were counted. The natural log of the proportion of wells without colonies was fitted linearly against the initial cell number in each well. The number of cells per resistance-initiating cell was calculated as $\frac{1}{1-e^{slope}}$ and the error of this value as $\frac{Error_{slope}\, e^{slope}}{(1-e^{slope})^2}$.

**Clone-tracing assay**. H3122 cells were transduced at 10% efficiency (as defined by fluorescence-activated cell sorting analysis of dsRed expression) with ClonTracer neutral DNA barcode library[21], kindly provided by Frank Stegmeier (Addgene #67267). Barcode containing cells were selected with puromycin and expanded for 23 days. Cells were expanded from $10^5$ cells (after puromycin selection) to $3.4 × 10^7$ cells. Quadruplicate cultures of $1.5 × 10^6$ cells each were plated into 10 cm dishes in the presence of 0.1% DMSO, 0.5 μM crizotinib, 0.5 μM alectinib, and 0.5 μM lorlatinib. Two aliquots of $1.5 × 10^6$ cells were frozen to serve as an initial mixture. Cells were collected following 4 weeks of cell culture. Genomic DNA was extracted using proteinase K digest, followed by phenol–chloroform purification. Barcodes were amplified, sequenced, and analyzed following protocols described in ref. [21] and an updated procedure provided on the Addgene website https://www.addgene.org/pooled-library/clontracer. A first round of amplifications was performed for 35 cycles using the primers 5′-TCG ATT AGT GAA CGG ATC TCG ACG-3′ and 5′-AAG TGG ATC TCT GCT GTC CCT G-3′ for 35 cycles. A second round was performed for 15 cycles using second-generation clonTracer primers with the following sequence: 5′-CAA GCA GAA GAC GGC ATA CGA GAT-Variable Sequence-GTG ACT GGA GTT CAG ACG TGT GCT CTT CCG ATC TCT AGC ACT AGC ATA GAG TGC GTA GCT-3′. Barcode enrichment analyses focused on barcodes that were enriched above the most frequent barcode within the average of the two baseline samples (0.0077). Heatmap.2 in R (using default parameters) was used for heatmap clustering. Corrplot[58] in R (using spearman correlation coefficients) was used to visualize correlations.

**Fluorescent in-situ hybridization analyses**. Cells were cultured in 10 cm dishes in 10 ml of 1640 medium supplemented with 10% FBS. After 24 h, 100 μl of colcemid (10 μg/ml, Life Technologies, Carlsbad, CA) was added to each culture and culture replaced to incubator for another 1 h before collecting. Cell suspension was transferred to 15 ml cortical tubes. Culture medium was removed by centrifuge. Cell pellets were subjected to hypotonic treatment with 0.075 M KCl and then fixed with Methanol and acidic acid at 3:1 v/v. Cell suspensions were dropped to slides and treated with 0.005% (wt/vol) pepsin solution for 10 min, followed by dehydration with 70%, 85%, and 100% (vol/vol) ethanol for 2 min each. Hybridization was performed by adding 10 μl of ALK dual-color break-apart probe on each slide (Cytocell, Cambridge UK), a coverslip was placed, and the slides were sealed with rubber cement. The specimens were subjected to denaturation at 75 °C for 3 min and hybridized at 37 °C for 16 h. The slides were washed in 0.4× saline-sodium citrate at pH 7.2 and then counterstained with 4′,6-diamidino-2-phenylindole (DAPI). Results were analyzed on a Leica DM 5500B fluorescent microscope. Cell images were captured in both interphase and metaphase cells.

**RNA-seq analysis**. Following 48 h of culturing in the absence of inhibitors (to minimize the impact of directly induced gene expression changes), RNA was isolated for erALK-TKI and parental H3122 cells using an RNAEasy Minikit (Qiagen). Reads were generated using a MiSeq instrument. Alignment was achieved using HiSat2 and the human hg19 reference genome. Normalized reads were obtained from DeSeq2. Analysis was performed with gene with more than 25 reads in at least one sample and more than two-fold change from ALK-TKI-naive cells in at least one resistant line.

**Nanostring assay**. Following 48 h of culturing in the absence of inhibitors (to minimize the impact of directly induced gene expression changes), RNA was isolated for erALK-TKI and parental H3122 cells using an RNAEasy Minikit (Qiagen). The nCounter GX Human Cancer Reference Kit was used to determine gene reads. Genes >1 SD from and >2-fold different from the mean parental value in any sample were retained for further analysis. The $\log_2\left(\frac{\text{Reads}}{\text{Median(Reads across all samples)}}\right)$ was used for all visualizations of the data. ClustVis[59] was used to generate PCA plots. Heatmap and corresponding dendograms were generated with heatmap.2 in R, using default parameters.

**SNP array**. DNA was extracted from parental and several biological replicates of ALKi-resistant H3122 cells using DNeasy Blood and Tissue Kit (Qiagen). DNA SNP sequencing was performed using CytoScanHD (ThermoFisher). Reads were processed using Normal Diploid Analysis in Chromosome Analysis Suite 3.2 (ChAS, ThermoFisher).

**Single-cell transcriptome analysis**. Single-cell expression was performed using 10× Genomics platform. Approximately 1000 cells were analyzed per each sample and ~40,000 reads per cell were generated using an Illumina NextSeq 500 instrument. Demultiplexing, barcode processing, alignment, gene counting was performed using the 10X Genomics CellRanger 2.0 software. Dimension reduction was done using UMAP[13], setting number of neighbors to 15. Cells were clustered using HDBSCAN on dimension reduced space[60], with the minimum number of cells set to 30. All other parameters were set to default values.

**Gene set enrichment analysis**. GSEA version 4.0 (ref. [61]) was used to determine the gene sets enriched in acquired resistant versus parental cell line RNA-seq profiles. We used the MSigDB Hallmark [PMID: 26771021] as the predefined gene sets and performed 10,000 permutations by gene set to determine the p-values. Gene sets with false discovery rate q-value ≤ 0.25 were considered as significantly enriched.

**ChIP-Seq analysis**. H3122 cells were fixed with 1% formaldehyde for 15 min and quenched with 0.125 M glycine. Chromatin was isolated by the addition of lysis buffer, followed by disruption with a Dounce homogenizer. Lysates were sonicated and the DNA sheared to an average length of 300–500 bp. Genomic DNA (Input) was prepared by treating aliquots of chromatin with RNase, proteinase K, and heat for de-crosslinking, followed by ethanol precipitation. Pellets were resuspended and the resulting DNA was quantified on a NanoDrop spectrophotometer. Extrapolation to the original chromatin volume allowed quantification of the total chromatin yield. An aliquot of chromatin (30 µg) was precleared with protein A agarose beads (Invitrogen). Genomic DNA regions of interest were isolated using 4 µg of antibody against H3K27Ac (Active Motif, catalog number 39133, Lot number 01518010). Complexes were washed, eluted from the beads with SDS buffer, and subjected to RNase and proteinase K treatment. Crosslinks were reversed by incubation overnight at 65 °C and ChIP DNA was purified by phenol–chloroform extraction and ethanol precipitation. Quantitative PCR (qPCR) reactions were carried out in triplicate on specific genomic regions using SYBR Green Supermix (BioRad). The resulting signals were normalized for primer efficiency by carrying out qPCR for each primer pair using Input DNA. Illumina sequencing libraries were prepared from the ChIP and Input DNAs by the standard consecutive enzymatic steps of end polishing, dA addition, and adaptor ligation. After a final PCR amplification step, the resulting DNA libraries were quantified and sequenced on Illumina's NextSeq 500 (75 nt reads, single end). Reads were aligned to the human genome (hg38) using the BWA algorithm (default settings). Duplicate reads were removed and only uniquely mapped reads (mapping quality ≥ 25) were used for further analysis. Alignments were extended in silico at their 3′-ends to a length of 200 bp, which is the average genomic fragment length in the size-selected library and assigned to 32-nt bins along the genome.

To define relevance toward transcriptional changes, RNA-seq reads were reanalyzed using HISAT2 and hg38 for proper comparison with ChIP-seq data. RseQC was used to check stranded information. HTSeq-count was used for reads counting. Raw ChIP-seq files were marked duplicate using picard and MACS2 was used to call peaks (q < 0.05). Peaks from four samples were merged using bedtools. Black list was removed. Featurecounts was used for read counting. All genes within ±20 kb of a peak are used for comparisons. Peaks with <15 reads in all samples with ChIP-seq and RNA-seq data (naive, erCriz0, and erAlec0, and erLor0) were excluded. Genes with <25 reads in all samples with ChIP-seq and RNA-seq data were excluded. Gene and peak levels were normalized using the DESeq2 rlog function. Genes with a difference in rlog values between erTKI and naive cells of ≥1 are considered upregulated. Genes with a difference in rlog values between erTKI and naive cells of ≤ −1 or less are considered downregulated. All other genes are considered neutral. Peak values that are associated with multiple genes are used multiple times.

*Tracks*: ACS2 was used to call peaks. Partek was used to visualize chromosomal tracks.

**Immunoblot analyses**. Protein expression was analyzed using NuPAGE gels (ThermoFisher), following the manufacturer's protocols. The following antibodies, purchased from Cell Signaling, were used: HER2 (4290), EGFR (2963), p-ALK Y1604 (3341), ALK (3633), p-Akt S473 (4060S), Akt (9272), pERK T202/T204 (4370S), ERK (4695), p-Stat3 Y705 (9145), Stat3 (9139), p-S6 S235/S236 (4858S), and s6 (2317) at 1:20,000 dilution. Anti-β-actin antibody was purchased from Santa Cruz (47778) and used at a 1:20,000 concentration. Secondary antibodies with H + L horseradish peroxidase (HRP) conjugates were purchased from BioRad (anti-rabbit: 170-6515, anti-mouse: 170-6516). HRP chemiluminescent substrate was purchased from Millipore (WBKLS0500). Images were taken using an Amersham Imager 600 (GE Healthcare Life Sciences).

**Xenografts studies**. erALK-TKI or parental H3122 were suspended in 1 : 1 RPMI/Matrigel (ThermoFisher) mix and subcutaneously implanted into 4–6-week-old NSG mice, with two contralateral injections per animal, containing $10^6$ tumor cells each. After 3 weeks, the animals were treated with 25 mg/kg alectinib (purchased from Astatech), 6 mg/kg lorlatinib (obtained from Pfizer), or vehicle control via daily oral gavage. Tumor diameters were measured weekly using electronic calipers and tumor volumes were calculated assuming spherical shaped tumors. Tumors in the collateral sensitivity experiments were treated with 11 mg/kg alectinib for 1 week before switching to 50 mg/kg alectinib. Xenograft studies were performed in accordance with the guidelines of the Institutional Animal Care and Use Committee of the H. Lee Moffitt Cancer Center. Animals were maintained under AAALAC-accredited specific pathogen-free housing vivarium and care and veterinary supervision following standard guidelines for temperature and humidity, with 12/12 light cycle.

**CRISPR-knockout experiments**. gRNAs were cloned into the pSpCas9(BB)-2A-Puro (PX459) V2.0 vector, using the protocol described in ref. [62]. The following oligo sequences were used: 5′-CAC CGT CTC TCG GAG GAA GGA CTT G-3′ with 5′-AAA CCA AGT CCT TCC TCC GAG AGA C-3′ and 5′-CAC CGC ATC CTG CTG GAG CTC ATG G-3′ with 5′-AAA CCC ATG AGC TCC AGC AGG ATG C-3′. H3122 cells were transfected with the above constructs using jetPrime reagent (polyPlus). Forty-eight hours following transduction, $10^5$ cells were seeded per 6 cm dish for a clonogenic assay. To account for differences in transfection efficiency between parental and resistant cell lines, control transfections were performed using MIG-GFP expressing plasmid and analyzed for percentage of GFP expressing cells using flow cytometry. Normalization was performed by multiplying the number of colonies by the transfection coefficient (ratio between fraction of GFP+ cells in the GFP control transfection between a given cell line and parental cells). pSpCas9(BB)-2A-Puro (PX459) V2.0 was a gift from Feng Zhang (Addgene plasmid # 62988; http://n2t.net/addgene:62988; RRID:Addgene_62988)[63].

**cDNA expression**. cMET overexpression was achieved using pLenti-MetGFP (Addgene #37560), a gift from David Rimm. The other entry cDNAs in pDO-NOR223 or pENTR221 were obtained from human ORFeome collection v5.1 or Life Technologies, respectively. Lentiviral expression constructs were generated by Gateway swap into pLenti6.3/V5-Dest vector (Life Technologies). Oncogenic fusion gene *EML4-ALK* variant 1 wild type or the *L1196M* point mutant in the retroviral pBabe-puro backbone[28] were provided by J. Heuckmann, (Universität zu Köln, Köln, Germany). Lentiviral and retroviral particles were produced, and were used for transduction of H3122 cells following standard protocols, as described in https://www.addgene.org/protocols/lentivirus-production/ and https://www.addgene.org/viral-vectors/retrovirus/retro-guide/.

**Flow cytometry analysis**. Tumors or lungs containing GFP/mCherry mixes were digested in RPMI supplemented with 1 mg/ml collagenase IV (Worthington), 1 mg/ml hyaluronidase (Sigma), and 2 mg/ml bovine serum albumin (Fisher Scientific) at 37 °C. Pellets were suspended in PBS and 0.1 µg/ml DAPI (Sigma). % mCherry+ and %GFP+ were determined using an LSRII flow cytometer (Backman Dickinson).

**Statistics and reproducibility**. Figures and statistical analyses of experimental data were performed using GraphPad Prism 8.3 software, R version 3.6.1, and Python arrays[64]. Statistical tests are stated in the figure legends. P-values < 0.05 were considered as statistically significant. Exact p-values are provided in Supplementary Table 2. Continuous variables were expressed as the mean ± SEM or mean ± SD as indicated in the figure legends. Unless otherwise stated, the experiments were performed at least three times with similar results. Single-cell transcription profiling, Nanostring, bulk sample RNA-seq, CRISPR knockout, and clone-tracing experiment were performed for a single time with biological replicates indicated in the figure legend and Methods section. Immunoblot analyses were performed two times with similar results. The original unprocessed and uncropped gels/blots with molecular weight marker information are shown in Supplementary Figs. 13–15 and provided as a Source Data file.

**Reporting summary**. Further information on research design is available in the Nature Research Reporting Summary linked to this article.

## Data availability

ChIP-seq data that support the findings of this study have been deposited in GEO with the accession code GSE144282. All other datasets during and/or analyzed during the current study are available from the corresponding author on reasonable request. ORFeome and MsigDB are publicly available.

## Code availability

Code is available on GitHub at nryoon12/Gradual_Development_of_ALK-TKI_Resistance.

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

## Acknowledgements

We thank M. Janiszewska, A. Goldman, and A. Rozhok for their critical reading of this manuscript and discussions, and I. Raplee for assistance with the RNA-Seq pipeline. This work was partially supported by Moffitt Lung Cancer Center of Excellence and the NIH (U54 administrative supplement 10-18279-04-13). D.L.P. was partially funded by an IMFAHE travel fellowship. J.G.S. is grateful to the NIH for their generous loan repayment program and to the Paul Calabresi Career Development Award for Clinical Oncology (NIH K12CA076917). This work has been supported in part by the Flow Cytometry Core, the Genomics Core and the Bioinformatics Core at the H. Lee Moffitt Cancer Center and Research Institute.

## Author contributions

A.M., R.V.V., and J.S. designed the studies. R.V.V., V.M., A.M., D.M., D.L.P., B.D., O.B., and L.K. performed experimental studies and analysed the data. N.Y. and A.D. developed mathematical models. R.V.V., A. Dhawan, A. Durmaz, J.P., M.T., M.A., O.M., and A.C.T. performed bioinformatical analyses. E.H. provided materials, clinical guidance, and suggestions of experimental and analysis approaches. M.A. reviewed the manuscript. A.M., R.V.V., N.Y., and J.S. wrote the manuscript.

## Competing interests

The authors declare no competing interests.
