## [Peer Review File · Nature Communications]

Reviewers' Comments:

Reviewer #1:

Remarks to the Author:

The authors have mostly addressed my comments. In the final version, the authors are urged to make it more clear the intention was not to uncover new resistance mechanisms but instead to consider the alternative model that resistance occurs via multi-factorial events gradually arising over time.

Reviewer #2:

Remarks to the Author:

I have been asked to comment on whether the concerns of reviewer #2 have been adequately addressed. The main concerns of reviewer #2 are that "evidence for their main conclusions is lacking. No additional experiments have been done to rule out other explanations for their observations. Therefore the manuscript continues to provide a collection of observations that are consistent with the authors' central thesis, but do not provide convincing evidence of its validity. I do not agree that the authors' items (1) through (5) provide sufficient evidence for resistance evolving as a gradual Darwinian adaptation."

My view is that, overall, the results reported in this manuscript are consistent with a more or less conventional framework: 1) Tolerance to modest drug pressure is relatively common among cancer cells and can be conferred by numerous molecular mechanisms; 2) Under modest drug pressure, drug-tolerant (persister) cells or their descendants accumulate compensatory adaptations that restore fitness; 3) One-hit mutations conferring complete resistance to high drug pressure are rare (<0.01% of cells) but are potentially clinically important because human tumours contain enormous numbers of cells. Therefore, although the manuscript contains novel and interesting results, I support reviewer #2's conclusion that it provides insufficient evidence to support a distinct new framework for understanding resistance to cancer drugs.

Specific comments:

Section II, lines 185-202: As the authors acknowledge (in lines 204-205), the results reported in this paragraph are consistent with the presence of drug tolerant (persister) cells, as reported in many previous studies (such as those cited on line 208).

Section II, lines 204-238: Given that similar, modest decreases in diversity (Fig S3a) and similar, modest subpopulation expansions (Fig S3b) were seen in both treatment and control conditions, it's unclear how this observation provides "Clear evidence of both negative and positive selection" as opposed to drift and/or measurement error. Also, since a "high degree of correlation between biological replicates" was seen not only within each treatment condition but also in the control condition, it is unclear how this constitutes evidence for "pre-existence of relatively stable weakly resistant subpopulations". The reason why barcode frequencies should be strongly correlated between control replicates is unclear. Perhaps the control (DMSO) conditions substantially differ from the ancestral conditions; perhaps the population was not yet well adapted to the ancestral conditions; or perhaps the barcodes incur fitness costs that vary according to integration site. In any case, since the treated populations resemble the controls, it's difficult to conclude anything about treatment effects from the results reported in these two paragraphs, except that the effects must be small.

Section III, lines 243-268: These results seem consistent with the presence of persister cells.

Section III, lines 291-316: Results of the agent-based model are consistent with resistance emerging from as few as three events (possibly two events, since apparently this scenario was not investigated). Evolution via only three (or perhaps two) events would not support a distinctive new paradigm of "gradual adaptation", as is the manuscript's main claim. Moreover, the agent-based model doesn't explicitly examine the scenario of reversible tolerance followed by compensatory adaptations, which remains a compelling, conventional explanation for the experimental observations.

Discussion, lines 579-582: I'm unconvinced by the claim that "This proposed framework essentially mirrors Darwin's original concept of evolution by natural selection". Such a claim requires supporting citations, especially given that scholars disagree about Darwin's views (e.g. see quotations and references in https://en.wikipedia.org/wiki/Phyletic_gradualism).

In case it might be of help to the authors, here I expand slightly on my previous comments.

The article claims (e.g. in lines 55-56, 84-89, 185-186, 547-556) to challenge a widespread view that resistance to cancer therapy occurs only via selection of pre-existing sub-populations with single-hit mutations. I think this is a straw man. Rather, I think it has long been generally understood that cancer cell populations can gradually adapt to moderate doses of cancer therapy in much the same way as to any other modest environmental change (such as in oxygen level, nutrients, growth factors, pH, etc.). In the particular case of very high therapeutic doses, the standard model invokes selection for very rare mutations conferring strong resistance. However, this study was unable to examine selection for very rare mutations because it looked at only small populations, relative to the size of human tumours. I therefore suggest withdrawing or rephrasing claims such as in lines 55-56, 84-89, 185-186, 547-556. Instead of claiming to overturn a supposedly widespread misconception, or to establish a new framework, I suggest the authors stick to explaining how more sophisticated understanding of the mechanisms and evolutionary dynamics of resistance may potentially benefit patients.

Below are some quotes from articles spanning three decades discussing ways in which resistance can occur. I'm not suggesting that the authors cite these particular instances. Rather, I mean to demonstrate that it isn't hard to find such articles.

"Taken together, this complex set of changes, observed in two very different models of broad resistance to xenobiotics, indicates that cells have the capacity to call upon an adaptive, coordinated defense mechanism when assaulted by cytotoxins. Once this program is turned on by one agent, it appears to be effective against others; in health such a plastic system may protect cells against environmental assault, but in disease it may protect neoplastic cells against chemotherapeutic agents."

Moscow, J. A., & Cowan, K. H. (1988). Multidrug Resistance. *JNCI Journal of the National Cancer Institute*, 80(1), 14–20. <https://doi.org/10.1093/jnci/80.1.14>

"Resistance is an adaptive phenomenon, involving a range of homeostatic responses on behalf of the normal and neoplastic tissues leading to a change in their relative proliferation rates. These changes occur over variable time periods following drug exposure, and some may show dose dependence. The initial response may be an attempt at detoxification either through metabolic inactivation of the drugs, or protection of critical sites from free-radical attack while later responses may be at the epigenetic or genetic level, including alterations in repair mechanisms, salvage pathways, drug efflux or interference with signal transduction. Selection pressure under certain tissue culture conditions may give prominence to one mechanism, and explain the disparity between clinical phenotype and experimental models. Somatic mutation is only one component in the host cell defence against cytotoxic drug attack."

Green, J. A. (1989). After Goldie-Coldman-Where now? *European Journal of Cancer and Clinical Oncology*, 25(5), 913–916. [https://doi.org/10.1016/0277-5379\(89\)90142-9](https://doi.org/10.1016/0277-5379(89)90142-9)

“Therefore, rather than an “all or none” phenomenon, resistance can be graded among cells in a population and, importantly, change in the same cell over time as it acclimatizes to environmental stresses. For example, PgP is a hypoxia inducible factor (HIF)1a client and its expression often increases in hypoxic and acidic environments even without cytotoxins. In addition, numerous mechanisms of de novo therapy resistance have been identified. For example, in environmentally mediated drug resistance (EMDR), components of tumor mesenchyma protect cancer cells from what would otherwise be lethal concentrations of cytotoxic drugs.”

Gatenby, R., & Brown, J. (2018). The Evolution and Ecology of Resistance in Cancer Therapy. *Cold Spring Harbor Perspectives in Medicine*, 8(3), a033415. <https://doi.org/10.1101/cshperspect.a033415>

“Sensitivity to drugs is often viewed as binary, where a cell that is exposed to drug simply dies if sensitive or survives if resistant. However, in reality tumors have a more nuanced mix of phenotypes. We investigated how various mixtures of sensitive and resistant cells compete to form a solid tumor mass, starting with cell cycle times randomly drawn from a normal distribution”

Gallaher, J. A., Enriquez-Navas, P. M., Luddy, K. A., Gatenby, R. A., & Anderson, A. R. A. (2018). Spatial Heterogeneity and Evolutionary Dynamics Modulate Time to Recurrence in Continuous and Adaptive Cancer Therapies. *Cancer Research*, 78(8), 2127–2139. <https://doi.org/10.1158/0008-5472.CAN-17-2649>

Response to the Critique.

We would like to thank the reviewers for the constructive criticism. Some of the characters in the reviewer comments, passed by the editor, did not display correctly. This did not impact the readability of the comments and we cite the reviewer comments in unedited form. Our responses are provided in blue font.

Reviewer #1 (Remarks to the Author):

The authors have mostly addressed my comments. In the final version, the authors are urged to make it more clear the intention was not to uncover new resistance mechanisms but instead to consider the alternative model that resistance occurs via multi-factorial events gradually arising over time.

We appreciate the reviewer comments and willingness to re-evaluated our revised submission. Whereas we did not explicitly claim, or intentionally implied, a discovery of new proximal mechanisms of resistance in our original submission, we make every attempt to avoid the possibility of misinterpretation regarding our claims of novelty of proximal resistance mechanisms in the revised manuscript.

Reviewer #3 (Remarks to the Author):

My view is that, overall, the results reported in this manuscript are consistent with a more or less conventional framework:

We appreciate the reviewer's time and effort to provide extensive feedback. We respectfully disagree about fit with the conventional framework. The field is rapidly developing with the wide introduction of single cell sequencing technologies, and a number of recent high-profile papers published after our initial submission report results consistent with our inferences. However, we would argue that the shift is not yet complete, and, based on highly cited pace-making reviews on the subject, as well as conference presentations and discussions, clinically-relevant resistance is typically attributed to either 1) pre-existence of fully resistant subpopulations, 2) single hit mutational transition of tolerant cells to *bona fide* resistance, 3) a drug-induced reprogramming process, or 4) cancer stem cell/EMT (which we view as explaining resistance away). Despite the emerging realization that these mechanisms are not necessarily mutually exclusive, the vast majority of clinical and experimental studies attribute clinically relevant resistance to a single mechanism, within the above categories. We think that our results/inferences, that point to multi-factorial resistance through acquisition of multiple mechanisms spanning different categories, are sufficiently novel, and we hope that this publication will contribute to the re-evaluation of the idea of single hit resistance.

1) Tolerance to modest drug pressure is relatively common among cancer cells and can be conferred by numerous molecular mechanisms;

We completely agree with the emerging commonality of tolerance (though it is still not universally accepted, especially within clinical and modeling communities), and we are not making claims of novelty for this phenomenon. From the expanded comments below, it appears that by "modest selective pressures" the reviewer implies clinical irrelevance. We would like to point out that the concentrations of ALK-TKIs used in this manuscript, in our unpublished follow up *in vivo* studies, as well as within several papers on resistance to other types of targeted therapies that we cite, use drug concentrations that are similar or higher than those

characterized *in vivo*¹. Therefore, we disagree with the implied message that the phenomenon of tolerance is only relevant to modest drug concentrations.

Whereas the emerging single cell based studies are revealing heterogeneity of tolerant phenotypes, we would argue that a more simplistic view of tolerance that reflects a single or a dominant mechanism is still very common – with several high profile papers, such as² making an argument that tolerance can be reduced to a single dominant mechanism, generalizable not only within a given tumor, but also across distinct types of cancers.

2) Under modest drug pressure, drug-tolerant (persister) cells or their descendants accumulate compensatory adaptations that restore fitness;

We obviously agree with the reviewer's assessment (with the qualification that the phenomenon is not limited to modest pressures). However, since the statement implies some sort of consensus, it is unclear what this consensus is based on. Following pioneering publications in EGFR mutant lung cancers^{3,4}, the idea of mutational conversion from tolerance to resistance is more or less commonly accepted, and it has been reproduced in multiple studies. However, we are not familiar with reports showing gradual accumulation of adaptations, apart from studies interpreting observations in light of reprogramming paradigm, such as in⁵.

3) One-hit mutations conferring complete resistance to high drug pressure are rare (<0.01% of cells) but are potentially clinically important because human tumours contain enormous numbers of cells. Therefore, although the manuscript contains novel and interesting results, I support reviewer #2's conclusion that it provides insufficient evidence to support a distinct new framework for understanding resistance to cancer drugs.

The numerical argument made by modeling studies (such as made in^{6,7}), and which we completely embraced prior to this study, is based on the widely accepted assumption that the single mutational hits are sufficient to provide full resistance. However, the evidence of 1-hit resistance is primarily based on studies that demonstrate the functional impact, but not sufficiency (such as in the commonly used BAF3 system). Our experimental studies with two distinct genetic mechanisms of resistance, which are commonly assumed to be sufficient to explain resistance and relapse directly challenge this assumption (Fig. 4), instead supporting additive action of multiple mechanisms, which can be acquired under continuous selective pressures. Obviously, experimental data in a single experimental system cannot completely refute the notion of pre-existing resistance of full resistance, stemming from a single mutational change. However, by challenging two poster-child cases of single hit resistance, our data at least warrants re-evaluation of the validity of the commonly accepted assumption of 1-hit mutational resistance.

Another important consideration, is that clinical relapse is often observed within tumors that show complete response (invisible to clinical imaging tools), with relapse occurring after months, and even years of therapy – which is arguably inconsistent with steady expansion of pre-existing subpopulations.

Finally, even rare fully resistant subpopulations pre-exist, under scenario of gradual development of resistance that we describe, these sub-populations would still be expected to compete with these gradually evolving cells – which could significantly alter the resulting evolutionary dynamics – a scenario that is not typically considered in the modeling community.

Specific comments:

Section II, lines 185-202: As the authors acknowledge (in lines 204-205), the results reported in this paragraph are consistent with the presence of drug tolerant (persister) cells, as reported in many previous studies (such as those cited on line 208).

We completely agree with the reviewer. We are not claiming the discovery of tolerance. Nor do we suggest that the idea of tolerance/persistence is irrelevant. We approached this project without an *a priori* assumption on gradual development of resistance, interpreting experimental data in light of existing paradigms (such as tolerance) for as long as reasonably possible, only proposing new interpretations to resolve the apparent discrepancies. We used the same logical sequence to present our studies in the manuscript. To address this and prior comments from the reviewer, we carefully evaluated and edited the text to make sure that our prose is sufficiently clear and does not contain phrasing that can be interpreted as implicit claims of discovering tolerance.

Section II, lines 204-238: Given that similar, modest decreases in diversity (Fig S3a) and similar, modest subpopulation expansions (Fig S3b) were seen in both treatment and control conditions, it's unclear how this observation provides "Clear evidence of both negative and positive selection" as opposed to drift and/or measurement error.

The high correlation within biological replicates of the same group is inconsistent with the idea of drift/measurement error; please note that all of the samples from the ClonTracer experiment were processed, sequenced and analyzed as a single batch. Direct comparison with the DMSO control is complicated by the big differences in the rates of cell proliferation, especially for the detection of positive selection. More modest differences in fitness over larger number of population doublings in the absence of ALK-TKIs might lead to the differences in barcode frequencies, comparable to those, resulting from much larger differences in fitness, but expanded over only a few cell doublings. At the same time, from the data reported in Figure 2G (which simultaneously presents data on all of the positively selected subpopulations), it is apparent that i) expansions are stronger and broader under ALK TKI selection, ii) the patterns of expansions are distinct – suggesting different selective pressures.

Interrogation of negative selection could potentially be more informative, but it would require substantial new developments in experimental and analyses pipelines, which we cannot afford without additional funding.

We agree with the reviewer that "clear evidence" might be an overstatement, thus "clear" was omitted.

Also, since a "high degree of correlation between biological replicates" was seen not only within each treatment condition but also in the control condition, it is unclear how this constitutes evidence for "pre-existence of relatively stable weakly resistant subpopulations".

Strong correlation of the expansion of specific barcoded subpopulations between multiple biological replicates, but not between different treatment conditions indicates selection of pre-existing subpopulations. We are not aware of any conceivable alternative explanation of the results. Also, please note that the approach of using selectively neutral barcodes to trace subclonal dynamics, and data analysis pipelines, have been used in multiple prior studies, such as^{8,9}, and to the best of our knowledge the methodology is not controversial. Of note, figure 2G shows that, despite a high degree of correlation, the expansion of these sub-populations is lower than in the DMSO control.

The reason why barcode frequencies should be strongly correlated between control replicates is unclear.

The observation of heterogeneity in the behavior of sub-populations within DMSO indicates (meta)stable differences in proliferation potential between barcode-labelled subpopulations. While we did not necessarily expect to observe this correlation, it is not entirely surprising either,

as many reports have described similarly complex subclonal behavior not only in vitro, but also in vivo (such as^{9, 10}). Some publications attribute it to stem cell dynamics¹⁰, but we are skeptical of this interpretation.

Perhaps the control (DMSO) conditions substantially differ from the ancestral conditions; perhaps the population was not yet well adapted to the ancestral conditions; or perhaps the barcodes incur fitness costs that vary according to integration site.

Since at 1:1000 dilution, used in these studies, DMSO does not impact cell proliferation, it is hard to imagine how it could alter proliferative dynamics. While continuous adaptation to the growth culture conditions and new fitness inequalities created by different integration size are more probable explanations, we think they are still unlikely to explain the observations. H3122 is a stable cell line that has been cultured for multiple generations since establishment and dissemination to different research groups. In multiple lines of experimental studies, we do not observe significant changes in proliferation rates through multiple passages under stable cell culture conditions (such as the same re-plating schedules, use of the same lot of serum etc). Given that there are apparent sub-groups marked with different barcodes, which behave near identical to each other, but differently under different drugs (**Figure 2G**), we think it is unlikely that the observed results reflect the impact of random integration. We know several research groups that try to make sense of this puzzling subclonal dynamic behavior, which is not limited to the use of exogenous barcodes, but has also been observed in cell lines where individual clones can be traced by heterogeneous immune receptor rearrangements (confidential personal communication). We are confident of the technical validity of the results and analyses, and we think that the reported results represent an interesting biological phenomenon. We think that this puzzle needs to be solved in order to properly understand subclonal behavior under drug-induced selective pressures (although we do not see an obvious approach to resolve it) – but it is outside of the main focus of our paper and does not directly impact our inferences.

In any case, since the treated populations resemble the controls, it's difficult to conclude anything about treatment effects from the results reported in these two paragraphs, except that the effects must be small.

We respectfully disagree with this assessment. Unsupervised clustering analysis shows that all three ALK TKI groups cluster distinctly from the control, and show higher expansion of these sub-populations (Figure 2G). As articulated above, unchallenged growth within the controls could lead to amplification of small differences in fitness over multiple population doublings, as opposed to bigger differences amplified with fewer cell divisions under the drugs.

Section III, lines 243-268: These results seem consistent with the presence of persister cells. We agree with this assessment. Please, note that the Clone Tracer experiment is meant to interrogate the pre-existence of sub-populations with different drug sensitivities, rather than saltational versus gradual mode of evolution.

Section III, lines 291-316: Results of the agent-based model are consistent with resistance emerging from as few as three events (possibly two events, since apparently this scenario was not investigated). Evolution via only three (or perhaps two) events would not support a distinctive new paradigm of "gradual adaptation", as is the manuscript's main claim. Moreover, the agent-based model doesn't explicitly examine the scenario of reversible tolerance followed by compensatory adaptations, which remains a compelling, conventional explanation for the experimental observations.

We agree with the reviewer that our *in silico* simulations results might be consistent with the emergence of resistance from tolerance from as little as two (epi) mutational events, as they clearly show consistency with three (epi) mutational events – under some of the mutational probabilities. Please note that the starting point in all of our simulations are tolerant cells i.e., cells within drug-naïve population that are capable of forming small colonies), and the scenario with bi-directional changes in fitness, consistent with reversibility of epimutational changes, was examined (**Fig. S4F**). As discussed above, we disagree that tolerance, followed by 2-3 additional mutational or epigenetic changes falls within a conventional framework. While several papers published after the initial submission support this idea, it is our assessment that the idea is still very far from being a convention – in the modeling, experimental and clinical communities.

Discussion, lines 579-582: Iâ€™m unconvinced by the claim that “This proposed framework essentially mirrors Darwin’s original concept of evolution by natural selection”. Such a claim requires supporting citations, especially given that scholars disagree about Darwin’s views (e.g. see quotations and references in https://en.wikipedia.org/wiki/Phyletic_gradualism).

We thank the reviewer for the suggestion. We have added a supporting quote from “On the Origin of Species”, “As natural selection acts solely by accumulating slight, successive, favourable variations, it can produce no great or sudden modifications”¹¹. We agree with the utility of supporting the claim (given the views on what constitutes a Darwinian paradigm within cancer research), but find it hard to imagine how one can disagree that Darwin proposed that evolution as a gradual process, since this is one of the central messages that he has articulated very explicitly numerous times, with many supporting examples. Whereas Darwin’s deep and complex writing might create openings for different interpretations, we think that the real issue is mis-interpretation of Darwinian paradigm reflective of lack of familiarity with his work, as, within the cancer research community, the term “Darwinian” is frequently used to express mutationalist views, once belonging to explicitly an anti-Darwinian camp.

On the point of the Wikipedia reference, we are familiar with the subject, and we have cited Gould & Eldredge as motivating some of our prior work¹². However, whereas punctuated equilibrium might have been at odds with views of some evolutionary biologists, the idea is not in disagreement with Darwin’s original views articulated in the Origin. Not only did Darwin not claim uniformity of evolution, he explicitly stated “..the periods during which species have undergone modification, though long as measured by years, have probably been short in comparison with the periods during which they retained the same form” – which seems to be perfectly consistent with the idea of punctuated equilibrium. On the other hand, Gould & Eldredge still considered gradual (rather than a single mutational hit) adaptation to a new equilibrium in a way that is consistent with Darwinian paradigm.

In case it might be of help to the authors, here I expand slightly on my previous comments.

We thank the reviewer for expanding on the previous comments, as this elaboration better explains the contention over our claims.

The article claims (e.g. in lines 55-56, 84-89, 185-186, 547-556) to challenge a widespread view that resistance to cancer therapy occurs only via selection of pre-existing sub-populations with single-hit mutations. I think this is a straw man. Rather, I think it has long been generally

understood that cancer cell populations can gradually adapt to moderate doses of cancer therapy in much the same way as to any other modest environmental change (such as in oxygen level, nutrients, growth factors, pH, etc.). In the particular case of very high therapeutic doses, the standard model invokes selection for very rare mutations conferring strong resistance. However, this study was unable to examine selection for very rare mutations because it looked at only small populations, relative to the size of human tumours. I therefore suggest withdrawing or rephrasing claims such as in lines 55-56, 84-89, 185-186, 547-556. Instead of claiming to overturn a supposedly widespread misconception, or to establish a new framework, I suggest the authors stick to explaining how more sophisticated understanding of the mechanisms and evolutionary dynamics of resistance may potentially benefit patients.

If we understand the comments correctly, the reviewer makes two major points here:

1. That adaptation to moderate drug concentrations can occur through gradual acquisition of multiple cooperating changes is a widely shared view – thus our argument is not novel.
2. Gradual development of resistance is irrelevant for clinics, as resistance to clinical concentrations of the drugs reflects selection of rare pre-existent variants conferring strong resistance – which our study has not rejected, by failing to consider very large populations, relevant to advanced multi-metastatic presentation at the onset of targeted therapies.

We would like to point out that we argue the relevance of gradual adaptation to high, clinically relevant or even higher drug concentrations, which we have used in our studies. Whereas our results cannot exclude the possibility of clinical resistance reflecting a selection of rare pre-existing subpopulations, we have experimentally interrogated two mechanisms, which are commonly referenced as mechanisms underlying clinical resistance: EML4-ALK amplification, and L1196M mutation – showing that these mechanisms alone do not provide a full degree of resistance.

That resistance reflects expansion of rare fully resistant sub-populations is hard to reconcile with long remission times (over a year), frequently observed in clinics with alectinib and other targeted therapies. This delayed relapse often occurs under continuous therapy from tumors that show complete response (dramatic reduction of population below the detectability threshold), and clinical relapse is often polyclonal. Yet, experimentally, we observe near-instantaneous relapse with tumor initiated with as little as 0.1% of fully resistant cells, (our unpublished observations).

Regarding the notion that it has been long understood that cancers can gradually adapt to drugs – while we have no reason to doubt that it might have been self-obvious to the reviewer, we have not encountered an articulation of this idea in reviews or discussion sections of primary research papers on the subject.

Below are some quotes from articles spanning three decades discussing ways in which resistance can occur. I am not suggesting that the authors cite these particular instances. Rather, I mean to demonstrate that it isn't hard to find such articles.

Taken together, this complex set of changes, observed in two very different models of broad resistance to xenobiotics, indicates that cells have the capacity to call upon an adaptive, coordinated defense mechanism when assaulted by cytotoxins. Once this program is turned on by one agent, it appears to be effective against others; in health such a plastic system may protect cells against environmental assault, but in disease it may protect neoplastic cells against chemotherapeutic agents.

Moscow, J. A., & Cowan, K. H. (1988). Multidrug Resistance. *JNCI Journal of the National Cancer Institute*, 80(1), 14–20. <https://doi.org/10.1093/jnci/80.1.14>

Resistance is an adaptive phenomenon, involving a range of homeostatic responses on behalf of the normal and neoplastic tissues leading to a change in their relative proliferation rates. These changes occur over variable time periods following drug exposure, and some may show dose dependence. The initial response may be an attempt at detoxification either through metabolic inactivation of the drugs, or protection of critical sites from free-radical attack while later responses may be at the epigenetic or genetic level, including alterations in repair mechanisms, salvage pathways, drug efflux or interference with signal transduction. Selection pressure under certain tissue culture conditions may give prominence to one mechanism, and explain the disparity between clinical phenotype and experimental models. Somatic mutation is only one component in the host cell defence against cytotoxic drug attack.

Green, J. A. (1989). After Goldie-Coldman-Where now? *European Journal of Cancer and Clinical Oncology*, 25(5), 913–916. [https://doi.org/10.1016/0277-5379\(89\)90142-9](https://doi.org/10.1016/0277-5379(89)90142-9)

Therefore, rather than an all or none phenomenon, resistance can be graded among cells in a population and, importantly, change in the same cell over time as it acclimatizes to environmental stresses. For example, PgP is a hypoxia inducible factor (HIF)1 α client and its expression often increases in hypoxic and acidic environments even without cytotoxins. In addition, numerous mechanisms of de novo therapy resistance have been identified. For example, in environmentally mediated drug resistance (EMDR), components of tumor mesenchyma protect cancer cells from what would otherwise be lethal concentrations of cytotoxic drugs.

Gatenby, R., & Brown, J. (2018). The Evolution and Ecology of Resistance in Cancer Therapy. *Cold Spring Harbor Perspectives in Medicine*, 8(3), a033415. <https://doi.org/10.1101/cshperspect.a033415>

Sensitivity to drugs is often viewed as binary, where a cell that is exposed to drug simply dies if sensitive or survives if resistant. However, in reality tumors have a more nuanced mix of phenotypes. We investigated how various mixtures of sensitive and resistant cells compete to form a solid tumor mass, starting with cell cycle times randomly drawn from a normal distribution.

Gallaher, J. A., Enriquez-Navas, P. M., Luddy, K. A., Gatenby, R. A., & Anderson, A. R. A. (2018). Spatial Heterogeneity and Evolutionary Dynamics Modulate Time to Recurrence in Continuous and Adaptive Cancer Therapies. *Cancer Research*, 78(8), 2127–2139. <https://doi.org/10.1158/0008-5472.CAN-17-2649>

We appreciate the reviewer for pointing us to these references, and highlighting specific passages. However, the relevance of these references towards our claims or reviewer points of contention are not clear to us.

The Moscow and Cowan review evokes an activation of a xenobiotic detoxification program, leading to multi-drug sensitivity. This reference is arguably relevant to the idea of resistance as reprogramming. It is not clear how this reference relates to gradual, multifactorial selection-driven acquisition of resistance. Moreover, given that different ALK inhibitors lead to different resistant phenotypes (despite a substantial cross-resistance), it is hard to reconcile our observations through a reprogramming paradigm, even if one ignores genomic changes that we observe.

The relevance of the opinion piece by Green is equally unclear. This well-articulated piece makes a case that appears to be relevant to the idea of acquisition of full resistance through tolerant intermediates (where tolerance is a cellular adaptation to drug induced stress). The paper makes some additional important points (including the importance of considering interactions with stroma), but we do not see a direct relevance to our arguments.

We are encouraged to see that references of Gatenby & Brown and Gallaher et al. are brought up. These are recent publications from the IMO department of Moffitt Cancer Center. Owing to the highly collegial academic environment, we have presented our study in multiple internal seminars (JS is a former IMO member, and, while AM is not formally a member of the department, he actively participates in seminars, discussions and events), and had multiple, extensive discussions about the evolution of drug resistance – which, hopefully, shaped the scientific views in both directions. We disagree that these publications capture the consensus in the field. At the same time, we still do not see a direct relevance of these citations to the main points that we are making in the manuscript.

Cited references:

1. Klempner, S.J. & Ou, S.H. Anaplastic lymphoma kinase inhibitors in brain metastases from ALK+ non-small cell lung cancer: hitting the target even in the CNS. *Chin Clin Oncol* **4**, 20 (2015).
2. Hangauer, M.J., Viswanathan, V.S., Ryan, M.J., Bole, D., Eaton, J.K., Matov, A., Galeas, J., Dhruv, H.D., Berens, M.E., Schreiber, S.L., McCormick, F. & McManus, M.T. Drug-tolerant persister cancer cells are vulnerable to GPX4 inhibition. *Nature* **551**, 247-250 (2017).
3. Hata, A.N., Niederst, M.J., Archibald, H.L., Gomez-Caraballo, M., Siddiqui, F.M., Mulvey, H.E., Maruvka, Y.E., Ji, F., Bhang, H.E., Krishnamurthy Radhakrishna, V., Siravegna, G., Hu, H., Raoof, S., Lockerman, E., Kalsy, A., Lee, D., Keating, C.L., Ruddy, D.A., Damon, L.J., Crystal, A.S., Costa, C., Piotrowska, Z., Bardelli, A., Iafrate, A.J., Sadreyev, R.I., Stegmeier, F., Getz, G., Sequist, L.V., Faber, A.C. & Engelman, J.A. Tumor cells can follow distinct evolutionary paths to become resistant to epidermal growth factor receptor inhibition. *Nat Med* **22**, 262-9 (2016).
4. Ramirez, M., Rajaram, S., Steininger, R.J., Osipchuk, D., Roth, M.A., Morinishi, L.S., Evans, L., Ji, W., Hsu, C.H., Thurley, K., Wei, S., Zhou, A., Koduru, P.R., Posner, B.A., Wu, L.F. & Altschuler, S.J. Diverse drug-resistance mechanisms can emerge from drug-tolerant cancer persister cells. *Nat Commun* **7**, 10690 (2016).
5. Shaffer, S.M., Dunagin, M.C., Torborg, S.R., Torre, E.A., Emert, B., Krepler, C., Beqiri, M., Sproesser, K., Brafford, P.A., Xiao, M., Eggan, E., Anastopoulos, I.N., Vargas-Garcia, C.A., Singh, A., Nathanson, K.L., Herlyn, M. & Raj, A. Rare cell variability and drug-induced reprogramming as a mode of cancer drug resistance. *Nature* **546**, 431-435 (2017).
6. Bozic, I. & Nowak, M.A. Timing and heterogeneity of mutations associated with drug resistance in metastatic cancers. *Proc Natl Acad Sci U S A* **111**, 15964-8 (2014).
7. Wodarz, D. & Komarova, N.L. Emergence and prevention of resistance against small molecule inhibitors. *Semin Cancer Biol* **15**, 506-14 (2005).
8. Bhang, H.E., Ruddy, D.A., Krishnamurthy Radhakrishna, V., Caushi, J.X., Zhao, R., Hims, M.M., Singh, A.P., Kao, I., Rakiec, D., Shaw, P., Balak, M., Raza, A., Ackley, E., Keen, N.,

- Schlabach, M.R., Palmer, M., Leary, R.J., Chiang, D.Y., Sellers, W.R., Michor, F., Cooke, V.G., Korn, J.M. & Stegmeier, F. Studying clonal dynamics in response to cancer therapy using high-complexity barcoding. *Nat Med* **21**, 440-8 (2015).
9. Lan, X., Jorg, D.J., Cavalli, F.M.G., Richards, L.M., Nguyen, L.V., Vanner, R.J., Guilhamon, P., Lee, L., Kushida, M.M., Pellacani, D., Park, N.I., Coutinho, F.J., Whetstone, H., Selvadurai, H.J., Che, C., Luu, B., Carles, A., Moksa, M., Rastegar, N., Head, R., Dolma, S., Prinos, P., Cusimano, M.D., Das, S., Bernstein, M., Arrowsmith, C.H., Mungall, A.J., Moore, R.A., Ma, Y., Gallo, M., Lupien, M., Pugh, T.J., Taylor, M.D., Hirst, M., Eaves, C.J., Simons, B.D. & Dirks, P.B. Fate mapping of human glioblastoma reveals an invariant stem cell hierarchy. *Nature* **549**, 227-232 (2017).
 10. Kreso, A., O'Brien, C.A., van Galen, P., Gan, O.I., Notta, F., Brown, A.M., Ng, K., Ma, J., Wienholds, E., Dunant, C., Pollett, A., Gallinger, S., McPherson, J., Mullighan, C.G., Shibata, D. & Dick, J.E. Variable clonal repopulation dynamics influence chemotherapy response in colorectal cancer. *Science* **339**, 543-8 (2013).
 11. Darwin, C. *On the Origin of Species by Means of Natural Selection, Or, The Preservation of Favoured Races in the Struggle for Life* (J. Murray, 1859).
 12. Marusyk, A. & DeGregori, J. Declining cellular fitness with age promotes cancer initiation by selecting for adaptive oncogenic mutations. *Biochim Biophys Acta* **1785**, 1-11 (2008).